



# An enhanced emissions module for the PALM model system 23.10 with application on PM$_{10}$ emission from urban domestic heating

Edward C. Chan[1,2], Ilona J. Jäkel[1], Basit Khan[3,4,5], Martijn Schaap[1,6], Timothy M. Butler[1,2], Renate Forkel[5], and Sabine Banzhaf[1]

[1]Institut für Meteorologie, Freie Universität Berlin, Germany
[2]Research Institute for Sustainability - Helmholtz Centre Potsdam (RIFS), Potsdam, Germany
[3]Mubadala Arabian Center for Climate and Environmental Sciences (ACCESS), New York University Abu Dhabi, United Arab Emirates
[4](Formerly) Institut für Photogrammetrie und Fernerkundung (IPF), Karlsruher Institut für Technologie (KIT), Karlsruhe, Germany
[5](Formerly) Institut für Meteorologie und Klimaforschung Atmosphärische Umweltforschung, Karlsruher Institut für Technologie (KIT), Garmisch-Partenkirchen, Germany
[6]Nederlandse Organisatie voor Toegepast-Natuurwetenschappelijk Onderzoek (TNO), Utrecht, The Netherlands

**Correspondence:** E.C. Chan (edward.chan@gfz-potsdam.de) and S. Banzhaf (sabine.banzhaf@met.fu-berlin.de)

**Abstract.** This article presents an enhanced emission module for the PALM model system, which collects discrete emission sources from different emission sectors and assigns them dynamically to the prognostic equations for specific pollutant species as volumetric source terms. Bidirectional lookup between each source location and cell index are maintained through using a hash key approach, while allowing all emission source modules to be conceived, developed and operated in a homogeneous

and mutually independent manner. An additional generic emission mode has also been implemented to allow the use of external emission data in simulation runs. Results from benchmark runs indicate a high level of performance and scalability. Subsequently, a module for modelling parametrized emissions from domestic heating is implemented under this framework, using the approach of building energy usage and temperature deficit as a generalized form of heating degree days. An model run has been executed under idealized conditions by considering solely dispersion of PM$_{10}$ from domestic heating sources.

The results demonstrate a strong overall dependence on the strength and clustering of individual sources, diurnal variation in domestic heat usage, as well as the temperature deficit between the ambient and the user-defined target temperature. Vertical transport contributes additionally to a rapid attenuation of daytime PM$_{10}$. Although urban topology plays a minor role on the pollutant concentrations at ground level, it has a relevant contribution to the vertical pollutant distribution.

**Keywords.** Urban scale modelling; Large-Eddy Simulation (LES); Domestic Emissions; Particulate Matter (PM); Algorithm

# 1 Introduction

Air pollution is one of the largest environmental and health risk factors in the European Union (EU; EEA, 2023). Despite ongoing efforts on improving general air quality, concentrations of airborne pollutants, such as particulate matter (PM$_{10}$), still frequently exceed EU standards across the EU-27 nations. It is particularly severe in urbanized areas, where 97% of the



population was exposed to PM at concentrations above guideline levels set forth by the World Health Organization in 2021
(EEA, 2023). Accordingly, a high population density also results in a larger variety of anthropogenic emission sources in
urban agglomerations. In addition to sectors in road traffic as well as industry and energy production, heat generation through
stationary combustion in residential and commercial buildings – collectively termed domestic heating – can be a significant
source of pollutants (Baumbach et al., 2010), constituting about 10% of urban emissions for nitrogen oxides ($NO_x$) and $PM_{10}$
(Senatsverwaltung Berlin, 2019; Pültz et al., 2023). To this end, urban-scale models such as the PALM model system (Maronga
et al., 2020) are indispensable tools for the evaluation of urban air quality. They can be deployed to assess the impact of different
emission reduction strategies or urban planning exercises for improving air quality (Jeanjean et al., 2017; Piroozmand et al.,
2020; Chan and Butler, 2021).

As emission data are often required as input to these models at high temporal and spatial resolutions (Guevara et al., 2020;
Chan et al., 2023), a suitable methodology for generating and handling such emissions data is critical for obtaining reliable
pollutant dispersion and transformation characteristics through numerical model runs at such scale. In the implementation of
the PALM model system (Maronga et al., 2020), emissions can only be modelled as boundary conditions on surface-bound grid
cells. Based in part on the existing surface module (Gehrke et al., 2021), time-varying emissions of a given pollutant species, for
example, particulate matter ($PM_{10}$) from vehicle traffic and domestic heating, are to be aggregated across all source locations.
They are then introduced into the solution domain as time-varying surface fluxes. This approach was used to provide hourly
traffic emissions, in which three different levels of detail (LOD) are available (Khan et al., 2021). Typically, emissions sources
are fully parametrized under LOD 0, through, for instance, redistribution of aggregate values according to rudimentary input
information for characterizing surrogate activity data (Gruney et al., 2017). On the other hand, the user has the option of
supplying already processed input emission data under LOD 2, independent of any predetermined parametrization schemes.
Partial parametrization, or LOD 1, is another available option where emissions are estimated from aggregated levels through
corresponding user-defined surrogate activity data.

Emission sources are typically categorized into sectors, exemplified by definitions in the Nomenclature for Reporting (NFR).
For conceptual generalization, different production mechanisms under each sector can be classified as such. However, additional flexibility should be provided to account for different emission source types (e.g., point, line, and area sources) as well
as physical mechanisms in pollutant formation within each sector. For example, $PM_{10}$ emissions from road traffic can be
originated from combustion products or from abrasion and resuspension sources, each requiring different physical parametric
treatment (Chan et al., 2023). Moreover, biogenic emissions such as isoprene (Guenther et al., 1991, 1993, 1999) and pollen
(Zink et al., 2012, 2013) contain different physical mechanisms and model treatments for emission release and replenishment.
Thus, the architecture and application of the emission sources in this framework should be organized in a similar fashion found
in prominent chemical transport models such as WRF-Chem (Grell et al., 2005) and LOTOS-EUROS (Manders et al., 2017).
On the other hand, as they ultimately contribute to the source and sink terms of the prognostic equation for the corresponding
species, the numerical construct of all emission sectors share certain common elements, in which abstractions can be drawn
across all relevant sectors to afford systemic uniformity and code reusability, significantly simplifying design, development
and deployment.





In addition, while the majority of emission sectors – such as traffic and domestic heating – are released as fluxes near
the surface i.e., ground or roof, this is not always the case. In particular, depending on the grid resolution, large emissions
sources – such as pollutant source terms from the European Pollutant Release and Transfer Register (E-PRTR) and point
source information from the Gridding Emission Tool for ArcGIS (GRETA; Schneider et al., 2016) – could be introduced over
a vertically distributed region about the respective industrial stack. Further, emissions from elevated sources – such as biogenic
emissions from trees and exhaust emissions from aviation – cannot be always be represented as surface fluxes at urban scales,
where grid resolution can be sufficiently high that they no longer take place on the first vertical layer alone. Conversely,
at sufficiently low horizontal resolutions, it becomes more likely that a given cell location could contain contributions from
multiple emission sectors. Although this could be partially addressed using the current surface flux based approach, in which
emissions are to be assembled *a priori* as surface flux boundary conditions, a yet greater degree of flexibility and independence
can be achieved if they could be introduced directly as volumetric source terms, where emission contributions could be specified
independently from each other at different points of time, and at different LODs.

The objective of this article is to introduce an enhanced emission module for the PALM model system, available from version
23.10 onwards. This module offers a high level of flexibility, implementation modularity, ease of use, and computational perfor-
mance. As an illustrative example, this new methodology is applied to parametrize emissions from domestic heating sources,
based on the energy demand approach of Baumbach et al. (2010) and Struschka and Li (2019). The numerical performance of
the enhanced emission module is evaluated using a synthetic test case executed at different levels of domain decomposition.
This is concluded by a demonstration in a residential region in Berlin under idealized conditions.

## 2 Model description

### 2.1 Theoretical foundations

Consider the incompressible prognostic equation at the resolved scale for the concentration of a given pollutant species $p$,
denoted as $\xi_p$, including net source term contributions:

$$\partial_t \xi_p + u_i \partial_i \xi_p - \Gamma_p \partial_j^2 \xi_p = \sum_r \sigma_p^r + \sum_m \epsilon_p^m \tag{1}$$

where $\Gamma_p$ is the diffusivity of species $p$, $\sigma_p^r$ is the corresponding net chemical conversion rate of species $p$ from all relevant
reactions $r$, and $\epsilon_p^m$ is the net emission rate of $p$ originating all relevant emission sectors $m$. The treatment of the $\sigma_p^r$ terms has
been thoroughly discussed in Khan et al. (2021); the present work thus concerns the incorporation of the $\epsilon_p$ terms into Equation
(1).

While the chemical production terms $\sigma$ are continuous in space and time, and are calculated everywhere in the computational
domain by solving a set of ordinary differential equations for temporal integration specific to a predetermined chemical kinetic
mechanism (Damian et al., 2002), $\epsilon$ are only defined at sparse, discrete regions and not always continuously active depending
on the emission sector. This represents, for instance, emissions from domestic heating through chimneys or road traffic. Since
emissions are typically supplied into the model as a mass or molar input rate (i.e., mol s$^{-1}$ or kg s$^{-1}$), provided either through





inventories (Jähn et al., 2020; Murphy et al., 2021) or parametrization (Huszár et al., 2010; Mues et al., 2014; Ge et al., 2020), converting these rates into concentration rates $\epsilon$ can be done by dividing the input rate with the corresponding mass or molar density at the computational cell where emissions take place. Therefore $\epsilon$ can be effectively regarded as a *volume source*. In terms of computer processing and storage, it is strongly preferred to consider $\epsilon_p$ only at discrete locations where the emission

source is present. However, there are three challenges associated from this approach, and the emission module architecture must be so conceived accommodate them in a sufficiently simple yet computationally robust and efficient manner:

1. As emissions sources are discrete, contiguity in the associated data structures (such as arrays) in the emission module no longer reflects to the spatial continuity in the computational domain,

2. The heterogeneity of emission sources from different sectors must be implicitly considered to enable different independent methods for parametrization and production mechanism specification, and

3. The interface between the prognostic equation solver and the emission module should be implemented to allow only localized data access to prevent propagation of data corruption into other emission sectors.

To address these issues, the spatial association between the individual emission source location and the corresponding $(i, j, k)$ cell index locations in the computational domain are maintained using a hash map ($h^m$) for each emission sector, where a hash

key $\kappa$ is assigned for each source location, that is:

$$h^m = \{\kappa^m(i,j,k) : \kappa^m \in W^{N_\kappa^m},$$
$$i \in W^{N_i^m}, \, j \in W^{N_j^m}, \, k \in W^{N_k^m}\}, \tag{2}$$

where $W$ denotes a set of natural numbers, i.e., $W \in 0, 1, 2, \cdots$ up to the corresponding upper bound $N$.

In a 3D Cartesian system, the hash key $\kappa$ can be derived from a linear mapping of the cell indices:

$$\kappa = N_i(kN_j + j) + i, \tag{3}$$

where $N_i$ and $N_j$ are the cell counts along the two horizontal axes in the computational domain. As $\kappa$ is unconditionally collision-free under this definition, a reverse lookup operation can be defined to recover the cell index location from $\kappa$ alone:

$$k = \mathrm{mod}\,(\kappa, N_i N_j), \tag{4}$$

$$j = \mathrm{div}\,[(\kappa - kN_i N_j), N_i], \tag{5}$$

$$i = \mathrm{mod}\,[(\kappa - kN_i N_j), N_i]. \tag{6}$$

It should be noted that Equations (4 - 6) must be performed in the order presented. Further, other methods for determining $\kappa$ exist, such as using different cell index ordering in Equation (3), or by using bitwise operations (Teschner et al., 2003).

Thus the volumetric emission source term can be expressed as the product between the source function $f_p^m$ for emission sector $m$ and $h^m$:

$$\epsilon_p^m = h^m \cdot f_p^m\left(t, \kappa^m | \eta_p^m\right), \tag{7}$$





where $f^m$ is a function of time ($t$), position (expressed succinctly in terms of $\kappa$), and additional sector-specific parameters ($\eta^m$) for pollutant species $p$ that are defined, for instance, when parametrization is involved. In turn, the transference of the modal emission sources for each sector to the linear system of discretized prognostic equations for pollutant species $p$ is the union all emission sectors:

$$\sum_m \epsilon_p^m = \bigcup_m \epsilon_p^m, \tag{8}$$

where $\bigcup$ is the union operator. In practice, the hash map belonging to each emission sector $m$ is amalgamated into a global hash map $H$, where:

$$H = \bigcup_m h^m, \tag{9}$$

and the source terms for each pollutant species $p$ under each sector $m$ is added directly to the corresponding species prognostic equations as required at each source location by looking up its $(i, j, k)$ cell indices from its hash key $\kappa$. This eliminates the need for intermediate data storage for source term accumulation.

## 2.2 Overall architecture

Figure 1 illustrates the overall architecture of the enhanced emissions module. Owing to different emission production mechanisms, emission source data ($\epsilon_p^m$) for pertinent pollutant species ($p$) originating each emission sector ($m$) are encapsulated in dedicated Fortran modules. Access of emission source data are restricted to standardized interface subroutines to prevent direct or accidental interference from other modules. Each mdoule under this framework also maintains a separate hash map ($h^m$) to indicate corresponding source locations represented by the hash key ($\kappa$). Beyond the physical mechanisms, all modules are similarly constructed, so that new emission sectors can be introduced easily.

The hash map for all emission sectors ($h^m$) will be merged to form a separate global hash map ($H$), which contains the ($i$, $j$, $k$) indices of all cells containing an emission volume source, as well as their respective hash keys ($\kappa$). The global hash map ($H$) and all associated subroutines and functions are contained in a separate module, whose interface is accessible by both the emission module and the prognostic equation solver. In this way, sector-specific emission sources ($\epsilon_p^m$) can be accumulated into the prognostic equation source terms at the correct cell locations. The global hash map also serves as a barrier between and the prognostic equation solver and the emission module under this architecture. As such, new emissions sectors can be developed without introducing code changes outside of the chemistry module in the PALM model system.

## 2.3 Implementation in the PALM model system

The enhanced emission module has been implemented and released for the PALM model system 23.10 (Maronga et al., 2020). This includes the parametrized domestic heating module described in Section 3. Both models are part of the chemistry module, and the interested reader is encouraged to refer to Khan et al. (2021) for further detail.





### 2.3.1 Program flow

Design decisions made on the overall architecture are based on user specifications on each LOD and active pollutant species. The LOD defines the extent of parametrization that takes place for an emission sector. Meanwhile, the emitting species relevant to the emission sector of interest can also be defined as input. Thus provisions should be given to render their treatment flexible. Each emission sector may operate in up to three levels of detail. Typically, LOD 0 indicates full parametrization, where all sector-specific parameters are defined solely in the namelist. On the other hand, in LOD 2, the user must provide all emission

source data covering the duration of the model run. Partial parametrization is referred to as LOD 1, where it requires both user emission data and namelist parameters as inputs. These sector- and LOD-specific parameters are analogous to the $\eta_p^m$ term in Equation (7). As input data, only user specified species that appear in the chemical mechanism will participate in the model run. Any species not defined by the user, or defined by the user but does not appear in the chemical mechanism, will be ignored. Species specification are to be performed independently for each emission sector.

Figure 2 shows the hierarchy of the new emission module within the PALM model system. The existing emission module is currently under the chemistry module (`chemistry_model_mod`), The new modules reside within the dotted line region, which include `chem_emis_vsrc_mod`, modules dedicated to each emission sector (`chem_emis_[sector]_mod`), as well as a module for the *generic mode* emission (`chem_emis_generic_mod`). Their roles in the enhanced emissions module will be detailed in the paragraphs below. Other modules that directly interact with the chemistry module are collectively referred

to as the *core*, and optional input data for each emission sector, that is, netCDF files required for user-defined emission data specification (otherwise known as level 2 of detail, or LOD 2), are not shown on Figure 2.

The module `chem_modules` contains definitions of all parameters that can be specified by the user in the namelist (`_p3d` file), and has been modified to contain activation and configuration options for individual emission sectors. Subroutine entry points for each emission sector are also introduced in the module `chemistry_model_mod` for initialization, as well as

emission source updates at specified intervals during the model run. The new module `chem_emis_vsrc_mod` contains the global hash map ($H$), where accumulation of prognostic equation source terms takes place through linkage to the hash maps of all activated emission sectors ($h^m$). The interface between `chemistry_model_mod` and the core modules in the PALM model system remains unchanged.

### 2.3.2 Emission module code structure

A structural overview of the Fortran modules for each emission sector ($m$) is presented in Figure 3. It comprises an interface to external modules and components; a data storage component for various sector-specific parameters such as LOD, active pollutant species information and hash key linkage to the prognostic equations; as well as methods (i.e., subroutines and functions) for general and LOD-specific operations and data manipulation. Data storage for each emission sector are kept private and can only be accessed through the publicly defined interface subroutines. This ensures encapsulation of modular

data and functionality. The interface serves as a wrapper to all internal functions and subroutines and are expected to be uniform for the emission module.





There are three subroutines defining the interface. First is initialization, which assigns user-defined data values specified in the namelist (`_p3d` file) as well as other optional external data sources. Throughout the model run, emission source values will be updated based on values at specific points of simulation time. This is done in the PALM model system core by calling the

update subroutine defined in the interface, which, in turn, invokes internal sector and LOD-specific subroutines for emission source calculation. A corresponding clean up function is called upon termination of the emission sector module to release all resources allocated during the model run.

The user can specify which sector(s) are to be activated for a model run and, if so, the corresponding LOD. The LOD-specific parameters, represented by the $\eta_p^m$ term in Equation (7), are also internal within each emission sector. The user can

also specify the pollutant species ($p$) to be used in the model run, which will be linked to the chemical species in the active chemical mechanism. Meanwhile, the hash map ($h^m$) contains information on all cell locations of emission sources through their hash keys ($\kappa$). Access between the individual source locations and the corresponding prognostic equation is established through the linkage between $h^m$ and the global hash map located in the module `chem_emis_vsrc_mod`, where all entries are sorted using an implementation of the `qsort` algorithm (Bentley and McIlroy, 1993) to facilitate the hash key lookup.

As mentioned in Section 2.3.1, the manner with which the emission module can be initialized, and how its emission sources are to be updated, are subject to the LOD. These specific subroutines and functions can be declared and implemented based on development specifications and requirements. In addition, implementation of various physical mechanisms and numerical constructs specific to the emission sector can be further abstracted into auxiliary subroutines and functions, which can be introduced at initial design or at subsequent development iterations as needed.

### 2.3.3 Generic emission mode

An additional generic mode has been introduced to provide an alternate possibility to provide emission source data for which an explicit emission sector is not (yet) available in the PALM model system. The generic emission mode contains no parametrization. Thus it is available only under LOD 2, in which the temporal-spatial data set generic emissions sources can be introduced into the model in a separate file in netCDF format. As such, the preparation and generation of these generic mode emission

data can be done outside of the PALM model system, thus maximizing user flexibility. Further, as LOD 2 data under various emission sectors are implemented in the same manner, the Fortran module for the generic emission mode also contains common functions and subroutines that can bed used in other emission sectors. These include but are not limited to user-defined and mechanism pollutant species matching, update interval detection, and basic data structure initialization and manipulation.

### 3 Parametrized emissions from domestic heating

As an illustrative example, underlying theory for parametrized (LOD 0) domestic heating, as well as its implementation under the framework described in 2.1, will be discussed. In addition, the test runs outlined in Sections 3.3 and 4 will also be based on this emission sector, from which computational performance and model results will be presented and accordingly discussed.





### 3.1 Emission source parametrization

The theoretical foundations for parametric modelling of domestic heating emissions are derived from the works of Baumbach
et al. (2010) and Struschka and Li (2019), which are based on a direct relationship between emissions and energy usage. Emissions are calculated using the so-called *emission factors*, which varies with the pollutant species and the furnace technology.
On the other hand, daily energy consumption are functions of the size, geometry, age, and function of the individual buildings.
These forms the two aspects of the discussion below. Typically, buildings with a footprint of less than 10 m$^2$ and a mean height
of less than 3 m are not considered in calculation.

The daily energy usage of a building ($E_B$) can be expressed as a ratio between its annual aggregate and the number of
degrees of temperature below the target temperature. In addition, diurnal variations due to general anthropogenic activity (i.e.,
heat tends to be turned up during early mornings and evenings, while turned down while sleeping) are also considered:

$$E_B\left(t\right) = \left(\left.\frac{E_B}{\Delta T}\right|_A\right)\zeta\Delta T(t), \tag{10}$$

where $E_B|_A$ is the annual energy consumption of the building $B$, $\Delta T|_A$ is the annually accumulated temperature deficit,
also known as heating degree, and $\Delta T(t)$ is the current temperature deficit, a generalized form of heating degree day (HDD)
to diurnal variations, and $\zeta$ is the diurnal variation in domestic heat usage, such as that defined by the Copernicus Atmosphere Monitoring Service (CAMS) for domestic and commercial combustion, under Gridded NFR (GNFR) sector C for other
stationary combustion (Kuenen et al., 2022).

The annual energy consumption ($E_B|_A$) takes into account the volume, compactness, energy demand of the building type
($\beta$), defined in Table B1:

$$E_B|_A = E_\beta|_A \, \Phi_\beta V_B, \tag{11}$$

where $\Phi_\beta$ is the compactness factor of the building type $\beta$ belonging to building $B$, $E_\beta|_A$ is the annual energy demand of the
building type $\beta$ per unit footprint area, and $V_B$ is the volume of the building $B$.

The compactness factor, in unit of m$^{-1}$, is a density indicator of the building. On the other hand, the annual energy demand,
in units of J m$^{-2}$ per annum, is the footprint-specific energy consumption. Tabulated values of these two quantities can be found
in Table B2 for each building type ($\beta$).

In turn, the temperature deficit $\Delta T(t)$ is calculated by subtracting the target indoor temperature ($T_0$) from the outdoor
ambient temperature ($T_\infty$):

$$\Delta T(t) = \max\left\{\, 0, [T_0 - T_\infty(t)]\,\right\}. \tag{12}$$

When the ambient temperature ($T_\infty$) is greater than the target temperature ($T_0$), there will be no temperature deficit (i.e.,
$\Delta T = 0$) and it is assumed that no heating will be required.

The volumetric emission for each species $p$ can then be calculated for each source location – represented by the corresponding hash key $\kappa$ – using Equation (7) in the function The function $f$, as a function of time $t$ and location $\kappa$. Here, the parameters,





i.e. $\eta$ in Equation (7), are represented in terms of the emission factor $\psi_p$ and building energy usage $E_B$, as shown in the relation

below:

$$f_p(t, \kappa \mid \psi_p, E_B) = \psi_p E_B, \tag{13}$$

where $\psi_p$ is the emission factor for the emitting species, whose values are tabulated in Table B3, and $E_B$ is the building energy usage determined in Equation (10). It should be noted that $\psi_p$ is a constant normalized by the building energy consumption ($E_B$) which, in turn, is a function of diurnal anthropogenic activities ($\zeta$) and temperature deficit ($\Delta T(t)$).

## 3.2 Module implementation

The domestic emissions module is implemented in the PALM model system under `chem_emis_domestic_mod`. In accordance with the overall architecture in Figures 2 and 3, interaction between the domestic emissions model and the PALM are made through the following interface subroutines:

- `chem_emis_domestic_init( )`,

- `chem_emis_domestic_update( )`, and

- `chem_emis_domestic_cleanup( )`,

which respectively handle LOD-specific module initialization, update of emission sources during the solver iterations, and relinquishing of allocated resources. The implementation of the fully parametrized (LOD 0) emission source term calculations are based on the formulation outlined in Section 2.1, and the user-defined emission source terms makes use of interface

functions and subroutines already defined in the generic emission mode described in Section 2.3.3. All options and parameters are defined in the module `chem_emission_mod`, and they can be specified by the user using the namelist (`_p3d` file).

The activation of the domestic emission parametrization can be specified by setting the namelist option `emis_domestic` to `.TRUE.`, while the corresponding LOD is specified using the option `emis_domestic_lod`. Currently, full parametrization (LOD 0) and user-specification (LOD 2) are available. Under LOD 2, the time series of all emission volume sources (that

is, their cell locations and the emission level) will be provided by the user explicitly through the input file with the suffix `_emis_domestic` in netCDF format, and no further namelist options are required. On the other hand, with LOD 0, additional parameters, presented in Table 1, can be specified by the user.

The emission source at each building stack location are updated at the start of the model run, and afterwards at every interval specified by the option `emis_domestic_update_interval`, which is 300 seconds by default. A user-defined

target temperature, with a default set to 15 °C, can be defined through the option `emis_domestic_base_temperature`, where domestic heating is assumed to be turned on when the ambient temperature falls below this target temperature. On the other hand, the option `emis_domestic_heating_degree` specifies the annual cumulative temperature (in degrees) to be heated above the ambient temperature to the target temperature, with a default value of 2100 K. The diurnal heat usage profile can be defined on an hourly basis via the option `hourly_diurnal_profile`, where the user can specify an hourly





weighting to represent aggregate anthropogenic activity for the region of interest. As default, the CAMS diurnal profile for residential and commercial combustion, defined under the GNFR sector C (other stationary combustion) is used (Kuenen et al., 2022). The compact factors and energy demands for each building type (see Appendix B) can be provided with the options `emis_domestic_compact_factors` and `emis_domestic_energy_demands`. The default values for both are defined in Table B2.

Meanwhile, emissions factors are to be presented in a name-value pair, where the species names are defined with the option `emis_domestic_species_names`. The corresponding emission factors are provided on the basis of unit energy consumed with the option `emis_domestic_species_emission_factors`. The emission factors for airborne species are to be presented in the unit of mol $TJ^{-1}$, and those for particulate species (such as $PM_{10}$) and other inert species are to be presented in kg $TJ^{-1}$. As the definition of chemical species in the PALM model system is mechanism-specific, the user must

supply the appropriate species names and emission factors, and is thus encouraged to refer to Table B3 for representative values of emission factors with different furnace technologies.

Further, for LOD 0, two additional variables defined in `_static` file are to be read as input. The first is `stack_building_volume`, which indicates the $(i, j)$ cell location of each building stack, and their corresponding building volume ($V_B$), typically assigned to each building belonging to a unique building ID. The second is `building_type`, which contains type of the building ($\beta$)

defined in Table B1, from which the compactness factor ($\Phi_\beta$) and energy demand ($E_\beta$), in Table B2 for example, can be found.

### 3.3  Performance benchmark

To provide an estimate of computational performance of the enhanced emission module, a synthetic test case a horizontal grid size of 400×400 cells has been created for evaluation. The test case domain contains 15 vertical layers with 129,600 uniformly distributed sources, representing 5.4% of the total cell count. Runs were conducted at three different levels of

horizontal domain decomposition: 10×10 processors (40×40 cells per compute core), 20×20 processors (20×20 cells per core), and 40×40 processors (10×10 cells per core). A simulation period of 3600 s is set for all cases at a fixed time step of 200 ms so that performance comparison across all runs can be made at a per time step basis. Mass conservation in the solution domain is verified by comparing the total input emission rate of an inert pollutant species (e.g., $PM_{10}$) into the solution domain to its total mass contained within in a cyclic lateral boundary arrangement. All runs are performed on the supercomputer

system hosted by the North German Association of High Performance Computing (Norddeutscher Verbund für Hoch- und Höchstleistungsrechnen; HLRN) using compute nodes comprising two Intel® Xeon® Platinum 9242 processors, totaling 96 cores, operating at a base frequency of 2.3 GHz, and 384 Gb of physical memory. The PALM model system has been compiled under `mpiifort` version 2018.6.288 with OpenMPI version 3.1.5.

At each domain decomposition level, control model runs are first conducted with the emission module deactivated, followed

by corresponding runs using the emission module. Thus the run time for the processing and solving of emission sources, which takes place every time step, can be calculated by taking the difference in the prognostic equation solver times between the domestic and reference runs. A set of 20 control and emission model runs are conducted for each domain decomposition level, and the runs with the two fastest and slowest prognostic equation solver time are discarded to attenuate outlier influence. Sum-





mary statistics are performed at both the aggregate level (i.e., all emission sources) and for a single emission source. Ultimately,
the per time step run times for processing an emission source and prognostic equation are calculated, with corresponding se-
rial performance data extrapolated from the three parallel runs, to evaluate the computational effort of the enhanced emission
module.

Table 2 shows the run time required to complete one time step of the prognostic equations for the control and emission
model runs at the three domain decomposition levels, along with results of statistical tests, based on results of 20 - 4 = 16
samples for each model run. To establish statistical significance in the difference in run times between the domestic and control
runs at each level of domain decomposition, inference in variance uniformity is first established by way of $f$-tests to determine
whether the pooled (statistically similar variance) or unpooled (statistically different variance) $t$-tests should be used. A level
of significance of 0.05 is used as guideline for all statistical tests.

The $f$-tests are performed using 16 - 1 = 15 degrees of freedom (DOF). From the results of the $f$-test, the difference in run
time are shown to be statistically significant for decomposition levels $10 \times 10$ (0.0159) and $40 \times 40$ (0.0167). On the other hand,
the difference at level $20 \times 20$ (0.207) is statistically inconclusive. The decision is thus made to use the unpooled treatment
for domestic and control run time distributions for the subsequent $t$-test, in which the effective DOF is calculated using the
Welch-Satterthwaite equation:

$$
\text{DOF}_{\text{effective}} = \frac{\left[\left(s^2/n\right)_{\text{domestic}} + \left(s^2/n\right)_{\text{control}}\right]^2}{\left[\frac{(s^2/n)^2}{n-1}\right]_{\text{domestic}} + \left[\frac{(s^2/n)^2}{n-1}\right]_{\text{control}}}, \tag{14}
$$

where $s^2$ is the unbiased variance estimator of each sample group (i.e., emission and control runs), and $n$ represents the
corresponding sample size (16 in both cases). The results from the $t$-test show that the difference in distributions for the run
times of the domestic and control runs are, indeed, statistically significant at all domain decomposition levels. This also means
the corresponding difference can be interpreted as the run time for the emission module. The difference in run time for each
decomposition setting were normalized by the total number of emission sources (129,600) to assess the numerical performance
of the domestic emission module and the volume source emission processing on a per stack or volume source basis for each time
step. This is presented in Figure 4, which can be linearized in a double logarithmic scale, from which the serial performance
can be estimated by way of linear regression. This is determined to be 1.3249 $\mu$s, representing the upper limit under the current
computing software and hardware configurations, with a coefficient of determination of 0.9946. This corresponds to a 0.512%
of serial processing time for all volume sources in the computational domain, where slight improvement under parallelization
can be seen, to 0.198% with $40 \times 40$ processors. Therefore, the benchmark demonstrates the effectiveness and scalability of the
present emission module, as well as other emission sectors utilizing the volume source based emissions processing framework.

## 4 Exemplary model run with parametrized emissions from domestic heating under idealized conditions

An idealized run case is conducted, following the performance benchmark, to assess the parametrized domestic emissions
module as an isolated emission source. The domain of this case study is located in a $800 \times 800$ m$^2$ cardinally aligned region
at the boarder between the districts Gesundbrunnen and Prenzlauer Berg of Berlin, at a horizontal resolution of 2 m, as shown



in Figure 5. The reference coordinate of the region is set to (52°32'32.6" N, 13°23'46.5" E) and is affixed to the origin of the computational domain. A reference elevation of 36.87 m above sea level is also introduced, from which the vertical ($z$) dimension of the computational domain extends to another 800 m above. A uniform grid spacing of 2 m was used in the $z$ direction, rendering the domain size of $400 \times 400 \times 400$ cells.

In the parametrization of domestic heating emissions, 228 out of 283 building units have met the minimum height (3 m) and footprint (10 m$^2$) criteria for which stacks are assigned. Emissions of inert $PM_{10}$ are parametrized for this idealized run. Reactive gas phase pollutants such as NO and $NO_2$, are not considered as their computed concentrations also depend on contributions of other reactive species from the background and other emission sectors not considered in this model configuration. It is also assumed that the pollutants emitted from the stacks are immediately well-mixed with the surrounding air in the col-

located grid cell. As such, the effects of micromixing resulting from segregation are not expected to be significant at such fine grid resolutions (Mastorakos, 2003; Gamory et al., 2009). The amount of pollutant species emitted from each domestic stack at any given time is calculated in accordance with the parametrization schemes set forth in Baumbach et al. (2010) and Struschka and Li (2019) described in Section 3.1. All buildings in the computational domain are assumed to be heated using a mixture of 50% centralized gas and 50 % oil furnaces (Table B3), according to the building technology data collected by the City of

Berlin for the region (Senatsverwaltung Berlin, 2010). The CAMS diurnal profile for residential and commercial combustion under GNFR sector C (Kuenen et al., 2022) is used for the calculation of the weighted temperature deficit profile ($\zeta \Delta T(t)$).

To further simplify the model, static profiles for meteorology have been used for initialization. A horizontal wind of ($u$, $v$) = (1.5, 0.5) m s$^{-1}$ is prescribed to provide constant bulk air movement. An initial temperature of 275 K with a vertical gradient of -0.1 K / 100 m is also introduced on the lateral boundaries to maintain a positive temperature deficit, $\Delta T(t)$, throughout the

run. A Dirichlet / Neumann boundary condition pair is applied for chemical species (i.e., $PM_{10}$) at each set of opposing lateral boundaries, while the Neumann boundary condition is applied to the top and bottom of boundaries of the solution domain. Furthermore, the following modules are also used to provide additional parametrization of relevant physical processes in the PALM model system under large-eddy simulation (LES) mode:

1. The land surface model (Gehrke et al., 2021) to solve energy balance of various surface types,

2. The plant canopy model (Maronga et al., 2020) for the parametrization of dynamic and thermodynamic processes of trees and vegetation,

3. The parametrized surface radiation scheme for calculating radiative energy budget under clear sky conditions (Maronga et al., 2020), and

4. The online chemistry module to model the dispersion of $PM_{10}$ (Khan et al., 2021).

It is worth mentioning that with the use of the clear sky radiation scheme, vertical divergence of the radiation fluxes leading to heating or cooling of the air column were excluded. In addition, as the purpose of the idealized model is to inspect the influence of anthropogenic emissions from domestic heating alone, other sectors of emissions, such as those from traffic and industrial sources, have also been precluded in this study.





The simulation has been set up for 48 hours, starting on 00:00:00 UTC on January 5, 2022, preceded by a 6-hour spin-up

period. Sampling takes place in the final 24 hours. The model run was performed on the HLRN supercomputer system described

in Section 3.3 on 20×20 compute cores. Time-averaged 3D output data (over a five-minute window) for all relevant prognostic

variables are written out at five minute intervals. Six sampling locations, representing various types of urban topology, have

been selected from the domain. Sampling location A represents a typical street canyon, where two long rows of buildings are

separated by a narrow road segment. Location B is situated in an open courtyard, that is, an open area surrounded by buildings.

The region immediately downwind of a large building is indicated in sampling location C, while a heavily built area but with

an open downwind region is represented by location D. Sampling location E is an open space in the middle of a park. Finally,

sampling location F is a completely closed courtyard. These locations are visually indicated in Figure 7 and tabulated in Table

4.

### 4.1   Temnperature deficit

To evaluate the behavior of the domestic heating parametrization throughout the idealized model run, the hourly mean nominal

domain temperature deficit $\Delta T(t)$ as well as the weighted deficit with the CAMS diurnal profile $\zeta \Delta T(t)$ are presented in

Figure 6. The temperature deficit varies with the ambient temperature. It begins at 00:00 UTC at 15.55°C and increases

monotonically, albeit slowly, to 15.78 °C at 07:00 UTC. Rapid changes follow in which the $\Delta T(t)$ drops to 7.89°C at 12:00

UTC, then returning to 13.6°C at 16:00 UTC. Towards the end of the day sees a mild increase, where $\Delta T(t)$ returns to 15.50°C

at the end. As $\Delta T(t)$ is positive throughout the day, the domestic stacks should be continuously operating and emitting $PM_{10}$,

in accordance with Equation (12).

On the other hand, the weighted temperature deficit $\zeta \Delta T(t)$, which exhibits a different behavior than the nominal $\Delta T(t)$,

which reflects the influence in meteorological conditions as well as anthropogenic activities on the expected energy usage and

the corresponding emission level, as indicated in Equation (10). Domestic heating is expected to be reduced substantially in

the nocturnal period from 01:00 to 06:00 UTC, where the majority of residents are at rest, and the weighted temperature deficit

hovers between 6.55°C (01:00 UTC) and 5.64°C (03:00 UTC). The morning peak takes place at 09:00 UTC, with an onset

starting at 08:00 UTC where $\zeta \Delta T(t)$ equals 18.9°C, but quickly reduces to 9.82°C at 13:00 UTC due to the corresponding

decrease in $\Delta T(t)$ in the daytime period. A steady recovery period follows, in which $\zeta \Delta T(t)$ rises to its evening peak value of

23.13°C, reflecting a generally high level anthropogenic activity, before dropping quickly to 15.50 °C at 00:00 UTC. While it is

impractical to draw trends from inspecting the output of all stacks due to variability in energy demands and local meteorology,

the weighted temperature deficit serves as a reasonable indicator on their overall level of operation, and by extension their

emission characteristics.

### 4.2   Spatial distribution of $PM_{10}$ concentration fields

Figure 7 shows the spatial distribution of diurnal mean $PM_{10}$ concentration evaluated at 2 m above ground. As expected, higher

$PM_{10}$ concentrations can be found in the wake of building clusters due to release of pollutants from the stacks. These can

range from approximately 2 to above 4 $\mu$g m$^{-3}$, depending on the location. On the other hand, in open areas, where stack





exhaust disperses into its surroundings, lower concentrations can be seen, to about 1 - 1.5 $\mu$g m$^{-3}$. This range of modelled concentration levels is in good agreement with the observed contributions of domestic heating of PM$_{10}$ concentrations found in existing studies for Berlin (Senatsverwaltung Berlin, 2019; Pültz et al., 2023).

The magnitude of the concentration also corresponds to the energy consumption ($E_B$) of the individual buildings, which depends on their volume and footprint according to Equation (11). Thus it can be seen that concentrations of PM$_{10}$ are higher following the wake of larger buildings (e.g., at locations B, C, and D). On the other hand, pollutant accumulation can be seen at locations where pollutant transport between the urban canopy and the free stream is restricted through the turbulent shear layer (Chan and Butler, 2021). Regions such as the street canyon (location A) as well as the closed courtyard (location F) are

exemplary of such pollutant accumulation. Particularly, the vicinity of location F contains a large number of stacks (Figure 5), which results in a large amount of local PM$_{10}$ being trapped into individual courtyards. In other cases, pollutants are transported from nearby buildings, in addition to local emission production, as qualitatively evidenced in the street canyon at location A.

### 4.3   Concentrations at sampling locations

The diurnal time series for the PM$_{10}$ concentrations at 2 m above ground for each sampling location is illustrated in Figure 8, in

which they follow a general trend. The concentrations reported from the idealized run are relatively low for all locations, with mean values between 1.65 and 3.11 $\mu$g m$^{-3}$ for each location. Higher concentrations can be observed towards the beginning and the end of the day, separated by a period during daytime, starting at 09:00 UTC, where the concentration is very low, averaging between 0.30 and 1.64 $\mu$g m$^{-3}$. As most locations (A to E) are positioned at some distance away from the emission sources at rooftop levels, the low concentrations indicate dilution through daytime vertical transport. This is not surprising,

considering that the prescribed wind for this idealized run is constant but low. Meanwhile, there is very weak dispersion during nighttime (from 00:00 to 08:00 UTC), which only causes the concentrations at all stations to decrease very slowly. The concentration recovers more quickly in the evening, coinciding an increase in $\zeta \Delta T(t)$. With the exception of locations C and F, where the recovery begins at 16:00 UTC.

    Location C is downstream of an isolated large emission source. Therefore it maintains a higher peak concentration than

the other stations, at 8.42 $\mu$g m$^{-3}$. On the other hand, without other emission sources in its proximity, the concentration at location C also takes longer to recover than at the other locations, at 19:00 UTC, but rises quickly to its evening peak (8.42 $\mu$g m$^{-3}$). By comparison, location F is in a closed courtyard in a large building complex. With numerous emission sources in its surroundings, a relatively high amount of PM$_{10}$ finds its way into the courtyard. However, pollutant exchange with the free stream flow is restricted through the rooftop shear layer (Chan and Butler, 2021), effectively trapping the PM$_{10}$ inside the

courtyard. This results in a steady increase of PM$_{10}$ concentration in the evening, to a peak of 9.49 $\mu$g m$^{-3}$. Following a sharp decrease at 01:20, corresponding to a decrease in anthropogenic activities, the concentration maintains at relatively constant level throughout the day, with a mean of 1.53 $\mu$g m$^{-3}$ between 02:00 and 16:00 UTC.

    Since domestic emissions are released from rooftops, further insights can be obtained by inspecting the vertical profiles of PM$_{10}$ concentrations at each sampling location, as illustrated in Figure 9. The profiles at four different points of time (08:00,

12:00, 16:00, and 20:00 UTC) are shown, representing the morning peak, daytime low, onset of recovery and evening peak





respectively. Due to the vertical positioning of the domestic stacks, $PM_{10}$ must be transported downwards to the ground level, which implies that lower concentrations are expected than those further up in the urban canopy, even in cases where the distribution is seemingly uniform (i.e., location F).

The concentration profiles themselves are also indicative of the urban topology of the corresponding sampling locations.
Location A is a typical street canyon, and a relatively stable vertical concentration profile can be found at all times. The canyon is well ventilated due to alignment of the road section with the wind flow. This results in a uniform vertical distribution of $PM_{10}$ concentration along and beyond the building height of 22 m on both sides of the canyon, with a mean of between 0.32 $\mu$g m$^{-3}$ at 16:00 UTC and 1.80 $\mu$g m$^{-3}$ at 20:00 UTC, to up to an elevation of 30 m above ground. As indicated in the diurnal mean concentrations in Figure 4, such stable distribution can be brought about by the emissions from the cluster of buildings about
300 m upstream from location A, as well as stacks originated from nearby buildings.

In contrast, location B is located in a cluster of buildings, and as such the vertical $PM_{10}$ concentration is dominated by emissions of local sources. It is also more vertically stratified than location A. The higher uniformity at 08:00 UTC, averaging 2.62 $\mu$g m$^{-3}$ up to the building height of 22 m, before diminishing near the rooftop. This indicates mixing of the $PM_{10}$ still lingering from the previous evening, which is almost completely dissipated by 12:00 UTC. At the onset of recovery (16:00
UTC), a peak at 2.92 $\mu$g m$^{-3}$ can be seen developing from the roof, and is being transported to ground level, as evidenced by the profile at 20:00 UTC, where peak concentration reaches 5.57 $\mu$g m$^{-3}$, while downward transport still takes place.

As previously mentioned, the concentration at location C is highly dependent on a single, large emission source located 185 m upstream. Therefore, while the concentrations could be significant, vertical mixing could be relatively weak, particularly due to the absence of other urban structures along the way to provide turbulence mixing and additional emission sources. This
single source can thus provide a level of concentration comparable to the other locations, to 4.72 $\mu$g m$^{-3}$ at the morning peak (08:00 UTC) and 6.05 $\mu$g m$^{-3}$ at the evening peak (20:00 UTC). However, for this reason it also is more vertically stratified. Further, at periods of low daytime emissions – indicated by $\zeta \Delta T(t)$ in Figure 6) – the $PM_{10}$ is seen to be totally dispersed before reaching location C.

Location D is situated in a similar topology as location B, where it is an open area surrounded by many buildings, with the
exception of a cluster of nearby buildings upwind, which also serves as a rich source of $PM_{10}$, as shown in Figure 5, particularly during the evening peak. Therefore, this contributes to the presence of uniform $PM_{10}$ even at higher elevations, to up to 2.32 $\mu$g m$^{-3}$ at 20:00 UTC. However, this uniformity is not maintained at the other points of time, suggesting the magnitude of emission as the primary driver for vertical transport. A similar observation can also be made for locations E and F, in which there are also numerous sources in their vicinity.

Despite being in an completely open area, the concentrations at location E is at a similar level as the other sampling locations, with maximum values of 2.51 $\mu$g m$^{-3}$ at 08:00 UTC and 3.71 $\mu$g m$^{-3}$ at 20:00 UTC. There is a strong dependence on contributions from nearby sources found in the building clusters at some distance upwind, as with location C. However, the $PM_{10}$ also diminishes quickly during periods of relative low emission levels (12:00 and 16:00 UTC) through dispersion.

The restriction of pollutant exchange at location F, compared to the other locations, has been prominently discussed the
previous paragraphs. The direct consequence of this is the accumulation of $PM_{10}$, which results in a highly uniform vertical





distribution at relatively high concentrations, with mean values ranging from 0.640 $\mu$g m$^{-3}$ at 16:00 UTC to 6.40 $\mu$g m$^{-3}$ at 20:00 UTC, up to the building height of 30 m. Mixing in higher altitudes is also evident, indicative of the contributions from numerous nearby emission sources upstream, as pointed out previously for locations D and E.

A qualitative representation of PM$_{10}$ dispersion can be seen in Figure 10. It shows the vertical distribution of PM$_{10}$ con-
centrations at each sampling location, indicated with triangular markers, from 08:00 to 20:00 UTC at four hour intervals. The sampling plane is taken at the north-south position of the corresponding location, spanning a width of 200 m in the east-west direction and covering an elevation of 100 m from the reference height. The higher concentration during peak periods have been discussed in Figure 9. Vertical mixing is also evident in each of the regions, albeit heavily attenuated during off peak times (12:00 and 16:00 UTC). The uniformity of PM$_{10}$ concentration inside the urban canopy can be seen in location B at
08:00 UTC, as well as location F at 20:00 UTC. The distribution plots provide a visual confirmation of the conclusions drawn from the previous figures. For instance, the much higher concentrations at the stack upstream of location C, which has been discussed previously, are clearly seen at 08:00 and 20:00. Moreover, contributions from nearby emissions can be observed at locations D and E, where relatively high levels of PM$_{10}$ reaches the sampling location from above, especially at 08:00 UTC.

## 5   Concluding remarks

An enhanced emission module has been developed for the PALM model system to provide a homogeneous operating framework for modelling emission contributions from various emission sectors at different levels of detail (LODs) and parametrizations. The hash key approach allows references from individual emission sectors to be made directly to the prognostic equation source terms in a highly effective and independent manner. In the mean time, modular interface functions and subroutines are provided to ascertain operational uniformity across all emission sectors with the PALM model system core, while encapsulat-
ing internal data and methods for each emission module and the framework itself. A benchmark run demonstrated excellent performance and scalability.

A module for domestic heating emission parametrization, based on the work of Baumbach et al. (2010) and Struschka and Li (2019) is implemented using the enhanced emission framework. The parametrization relates emissions to energy demand of each individual building, which is driven by the building topology and the deficit between the ambient temperature and the
target indoor temperature, weighted by the diurnal variation in domestic heat usage. In addition, the stack emissions output are influenced by predetermined tabulated factors for different building types and age, as well as emission factors for different pollutants and furnace technologies, which are used in the parametrized module as default values.

Subsequently, an test case under idealized conditions is conducted for the parametrization module is conducted over a 800 × 800 m$^2$ residential region in Berlin. The model run covers a 24-hour sampling period following spin-up under constant
wind and surface temperature. The diurnal time series for nominal ($\Delta T(t)$) and weighted ($\zeta \Delta T(t)$) temperature deficit reflects continuous furnace operation with highly varying energy usage and emission output throughout the day, showing a short day time peak and a longer evening peak. Near surface PM$_{10}$ concentrations at different sampling locations show similar trends, showing very low levels during daytime due to low levels of heating and dispersion through vertical transport. Meanwhile,





values are persistently moderate during nighttime following the evening peak. Vertical concentration distributions vary de-
pending on topology of each sampling location. It is shown that a large number of nearby emission sources heavily influences
vertical distribution at elevations beyond building height, while the uniformity of the vertical column is dependent on the level
of ventilation of the area surrounding each location.

The performance and functionalities of the new approach in the PALM model system have been fully demonstrated in
the present work through the benchmark (Section 3.3) and the idealized run (Section 4). This approach to emission source
treatment enables emission models for the PALM model system to be designed and developed in a highly intuitive, flexible and
independent manner. For the end user, the modularization of different emission sectors offers a higher degree of control over
execution options and details, allowing studies to be conducted with improved organization and precision. Finally, a built-in
generic emission mode gives the user the ability to import third-party emission data for sectors not yet available in the model
system.

In addition to the domestic emission parametrization module featured in this work, implementation of additional modules
for the parametrization of biogenic VOC emissions, pollen dispersion, and E-PRTR or GRETA (Schneider et al., 2016) point
sources are available from the most recent release (23.10) of the PALM model system at the time of writing. Further enhance-
ments have been planned for the parametrized domestic emission module itself. These include building heat loss estimation
and thermal effects on stack exhaust dispersion. Finally, a simplified characterization of ground-based emission sources (such
as road traffic) can be represented as surface fluxes (Khan et al., 2021; Gehrke et al., 2021) within the present emission module
framework.

## Appendix A: Nomenclature

### A1 Roman Symbols

| | |
|---|---|
| $A$ | Annual aggregate (subscript) |
| $B$ | Building index (subscript) |
| $E_B$ | Building energy consumption [kW h] |
| $E_\beta$ | Per footprint energy demand for building type $\beta$ [kW h m$^{-2}$ p.a.] |
| $f$ | Parametrized emission source function in Eqns. (7) and (13) |
| $H$ | Amalgamated global hash map |
| $h^m$ | Hash map for emission sector $m$ |
| $i, j, k$ | Cell indices |
| $m$ | Emission sector index (superscript) |
| $n$ | Sample size in Eq (14) [ ] |
| $N$ | Array (vector) size |
| $p$ | Index for pollutant species (subscript) |

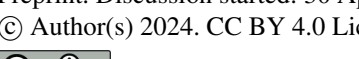



| $r$ | Index for chemical reaction (superscript) |
|---|---|
| $s^2$ | Unbiased sample variance estimator [$\cdot^2$] |
| $T$ | Temperature [K] |
| $T_0$ | Base building temperature [K] |
| $T_\infty$ | Ambient temperature [K] |
| $\Delta T$ | Temperature deficit [°C] |
| $\Delta T_A$ | Annual heating degrees [°C p.a.] |
| $t$ | Time [s] |
| $u$ | Wind velocity [m s$^{-1}$] |
| $V$ | Building volume [m$^3$] |
| $W$ | Natural number ($\in 0, 1, 2, \cdots$) |

## A2   Greek symbols

| $\beta$ | Building type index (subscript) |
|---|---|
| $\epsilon$ | Volumetric emission rate [m$^{-3}$ s$^{-1}$] |
| $\zeta$ | Diurnal weight factor for anthropogenic activity [ ] |
| $\eta$ | Generalized parameter in Equation (7) |
| $\Gamma$ | Species diffusivity [m$^2$ s$^{-1}$] |
| $\kappa$ | Hash key |
| $\xi$ | Pollutant concentration [$\cdot$ m$^{-3}$] |
| $\sigma$ | Chemical production rate [m$^{-3}$ s$^{-1}$] |
| $\Phi$ | Building compactness factor [m$^{-1}$] |
| $\Psi$ | Emission factor [mol TJ$^{-1}$ or kg TJ$^{-1}$] |

## A3   Acronyms and abbreviations

| CAMS | Copernicus atmosphere monitoring service |
|---|---|
| DOF | Degree(s) of freedom |
| E-PRTR | European pollutant release and transfer register |
| (G)NFR | (Gridded) nomenclature for reporting |
| GRETA | Gridding emission tool for ArcGIS |
| HDD | Heating degree day |
| HLRN | (Norddeutscher Verbund für) Hoch- und Höchstleistungsrechnen |





| KPP | Kinetic PreProcessor (Damian et al., 2002) |
| LES | Large eddy simulation |
| LOD | Level of detail |
| $PM_{10}$ | Particulate matter with mean diameter of up to 10 $\mu$m |
| VOC | Volatile organic compound |

## Appendix B: Building-Specific Parameters for Domestic Heating Emissions

Six building types are supported in the current implementation of parametrized domestic heating emission mode, based on their function and construction period, to represent their energy consumption characteristics, as presented in Table B1.

Each building type ($\beta$) is further defined by its compactness factor ($\Phi_\beta$) and annual energy demand ($E_\beta$). The following results are taken from Struschka and Li (2019) and are presented in Table B2 below.

Further, Struschka and Li (2019) has also proposed emission factors for carbon monoxide (CO), oxides of nitrogen ($NO_x$ and $NO_2$), particulate matter ($PM_{1010}$) and volatile organic compounds (VOC). These values have been tabulated in the table below according to each domestic furnace technology (Table B3). Please note that the emissions factors used for the idealized run in Section 4 are taken as the arithmetic mean between centralized oil and centralized gas to reflect the furnace technology composition of the region of interest (Senatsverwaltung Berlin, 2010).



*Code and data availability.* The exact version of the source code for the PALM model system containing the featured emission module is licensed under the terms of the GNU General License version 3.0 or later and can be obtained using the digital object identifier 10.5281/zenodo.10890465. A user guide on building the featured module from source, as well as executing the accompanied test cases, are located in the supplement indicated below. Additional information on available input options for the parameterized domestic emissions module and their
applicable nominal values can be found in Section 3.2 and Appendix B.

*Supplement.* The supplement related to this article can be obtained using the digital object identifier [DOI].

*Author contributions.* Initial conception was carried out by ECC, SB, BK, and RF. Theoretical foundations for domestic emissions were laid out by IJJ, SB, and ECC. ECC assumed lead design and development with contributions from IJJ and RF. ECC and BK assembled and executed all model runs. The manuscript and all supporting data were prepared by ECC, IJJ, BK, and SB. TMB and MS provided technical
and logistic guidance at all stages of the study.

*Competing interests.* The contact author has declared that none of the authors has any competing interests.

*Acknowledgements.* Approaches on module architecture and integration with the PALM model system, particularly its LES dynamic core and chemistry solver, were put forward following thorough discussions with Tobias Gronemeier (iMA Richter & Röckle GmbH, Freiburg im Breisgau, Germany) and Matthias Sühring (Leibniz Universität Hannover). Theoretical foundations concerning parametrization of domestic
heating emissions were clarified in detail by Michael Struschka. Aurelia Lupaşcu (ECMWF, Bonn, Germany) and Konstantinos Michos (Winterthur Gas & Diesel Ltd., Switzerland) provided valuable input on language-specific implementation. Siegfried Raasch (Leibniz Universität Hannover) was responsible for the overall project leadership and coordination, as well as the maintenance and technical support of the PALM model system repository. All computations have been performed on high performance clusters maintained by the HLRN.

*Financial support.* The work outlined in this article is supported in part by the funding instrument "Urban Climate Under Change" (Grant
ID: 01LP1911I) from the German Federal Ministry of Education and Research (New ; BMBF).





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





**Table 1.** Namelist options for domestic emissions parametrization (LOD 0).

| Option (`emis_domestic_ ...`) | Term | Units |
|---|---|---|
| `update_interval` | -- | s |
| `base_temperature` | $T_0$ | °C |
| `heating_degree` | $\Delta T_A$ | K |
| `hourly_diurnal_profile` | $\zeta$ | -- |
| `compact_factors` | $\Phi_\beta$ | m$^{-1}$ |
| `energy_demands` | $E_\beta$ | kW h m$^{-2}$ p.a. |
| `species_names` | $p$ | -- |
| `species_emission_factors` | $\Psi_p$ | (mol or kg) TJ$^{-1}$ |

**Table 2.** Per time step mean [ms] and variance [ms$^2$] for prognostic solver run time for control runs and emission module runs at each domain decomposition level, along with effective DOF [ ] from Eq. (14), statistical significance [ ] for variance uniformity (*f*-test) and sample difference (*t*-test). The number of samples is 16 for all runs, and the level of significance is set to 0.05.

| | Control runs | | Emission runs | | Difference | | Statistical significance | | |
|---|---|---|---|---|---|---|---|---|---|
| Domain | Mean | Variance | Mean | Variance | Mean | Variance | DOF | *f*-Test | *t*-Test |
| 10×10 | 221.77 | 6.4071E-2 | 222.37 | 2.0139E-2 | 0.60126 | 5.2631E-3 | 23.582 | 1.5863E-2 | 2.3265E-8 |
| 20×20 | 42.596 | 3.0809E-4 | 42.734 | 4.7399E-4 | 0.13887 | 4.8880E-5 | 28.708 | 2.0687E-1 | 4.8546E-18 |
| 40×40 | 10.531 | 2.0162E-4 | 10.552 | 6.4157E-5 | 0.02086 | 1.6611E-5 | 23.668 | 1.6744E-2 | 3.4813E-5 |

**Table 3.** Stack processing time and total solver run time per time step for each emission volume source.

| Domain | Serial | 10×10 | 20×20 | 40×40 |
|---|---|---|---|---|
| Stack [$\mu$s] | 1.3249 | 4.6394E-3 | 1.0716E-3 | 1.6096E-4 |
| Solver [$\mu$s] | 258.94 | 1.7112 | 0.32867 | 8.1257E-2 |
| Performance [%] | 0.51167 | 0.27112 | 0.32603 | 0.19808 |





**Table 4.** Sampling locations in the computational domain (Figure 5). Easting and northing coordinate values are relative to the domain origin (52°32'32.6" N, 13°23'46.5" E).

|   | Coordinates [m] | Description |
|---|---|---|
| A | (460, 680) | Street canyon |
| B | (90, 230) | Open courtyard |
| C | (740, 300) | Behind large building |
| D | (700, 610) | Semi open space |
| E | (300, 450) | Park |
| F | (488, 514) | Closed courtyard |



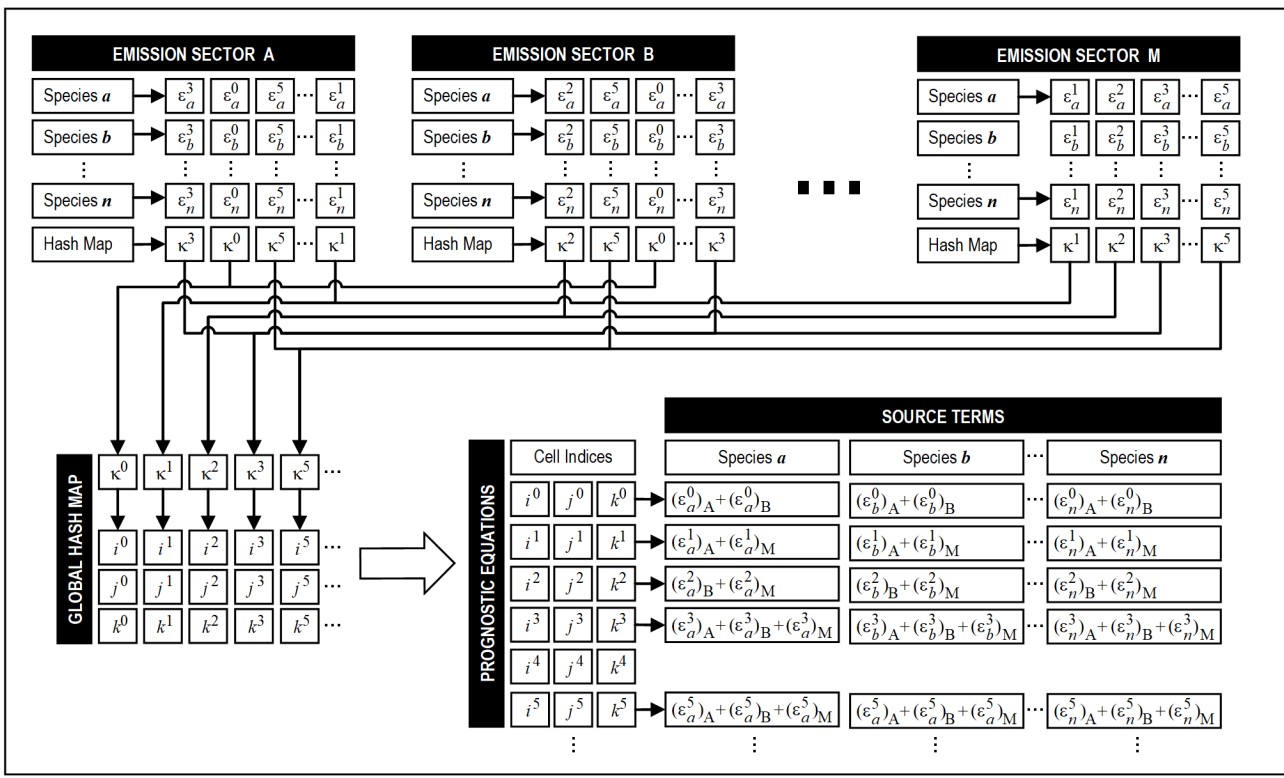

**Figure 1.** Overall architecture of the enhanced emission module in the PALM model system. Chemical species are indicated in lower case letters ($a, b, c, ...$) and emission sectors are in upper case letters ($A, B, C, ...$). Both $n$ and $M$ respectively denote an arbitrary number of species and emission sectors. Hash keys ($\kappa$) and corresponding cell indices ($i, j, k$) are distinguished by superscript numerals.



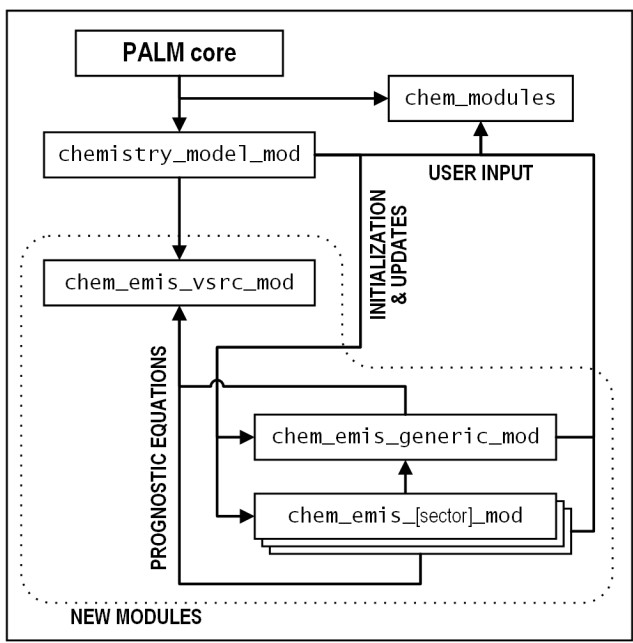

**Figure 2.** Hierarchy of the new source files (inside the dotted line region) introduced for the enhanced emission module of the the PALM model system. Arrows indicate direction of module access.





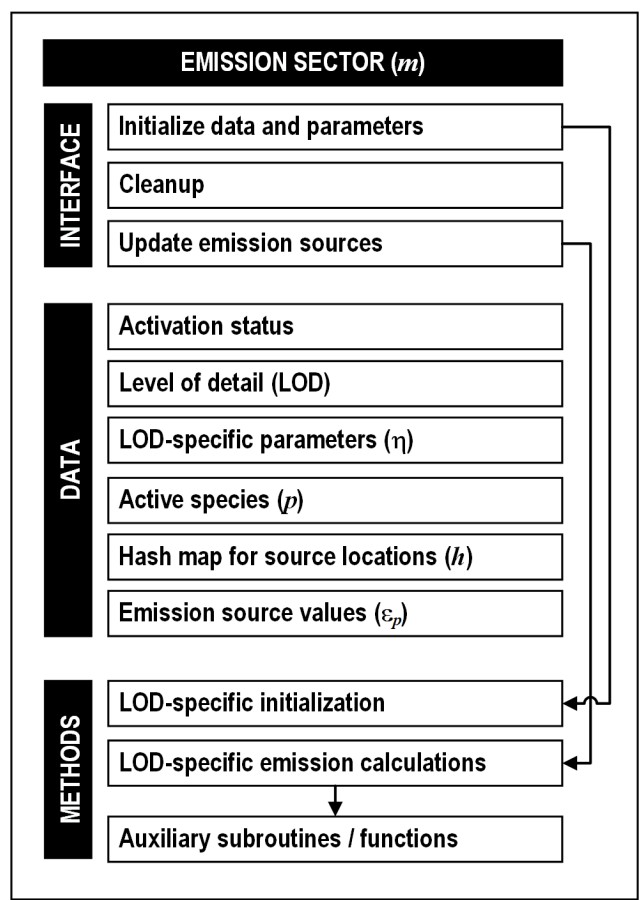

**Figure 3.** Overall structure of a module for an emission sector under the enhanced emission framework. Module data and methods are encapsulated, to be accessed externally through the use interface subroutines.



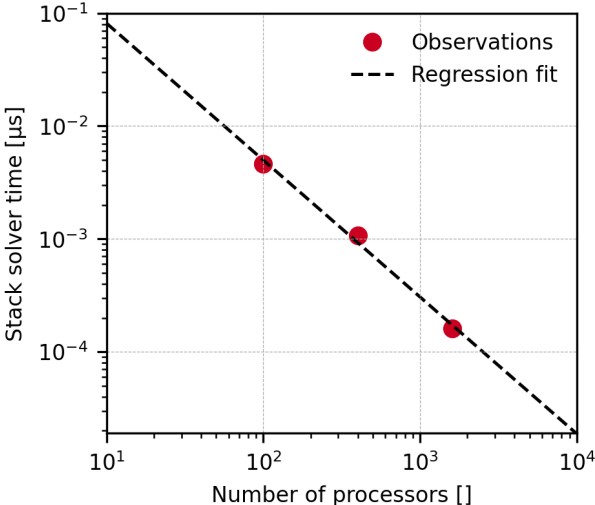

**Figure 4.** Per time step stack solver run time at different domain decomposition levels.





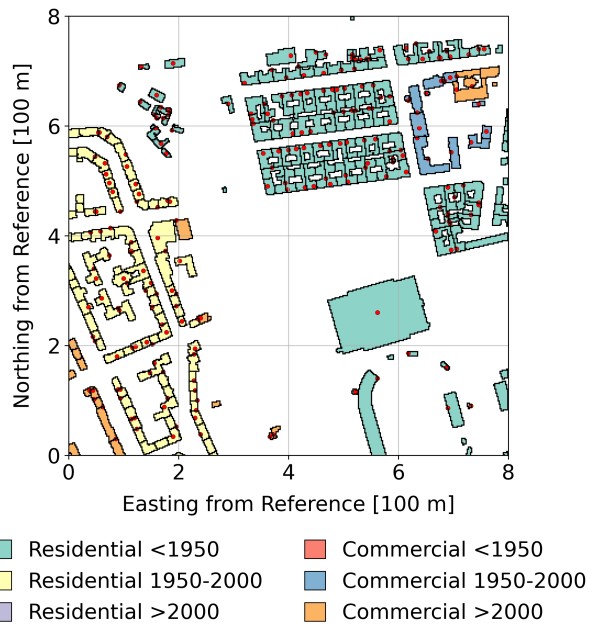

**Figure 5.** Horizontal distribution of urban structures residing in the region of interest for the idealized model run. Color indicates the building type (Table B1). The locations of chimney stacks are shown in red circular markers. The domain origin is set to (52°32'32.6" N, 13°23'46.5" E).



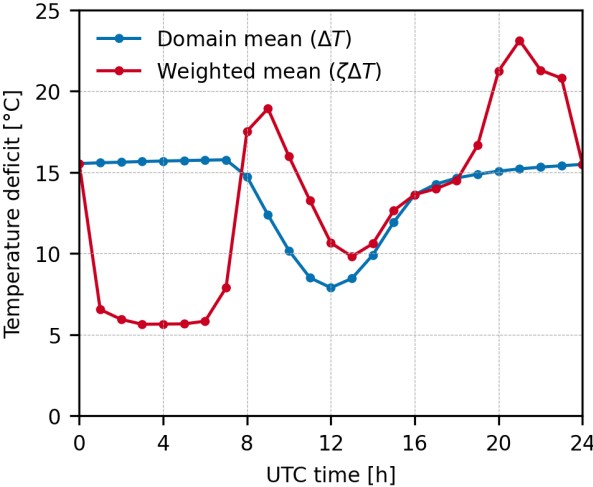

**Figure 6.** Domain hourly mean for temperature deficit (blue) and hourly temperature deficit weighted by CAMS diurnal profile (red) for residential and commercial combustion obtained (under GNFR sector C).



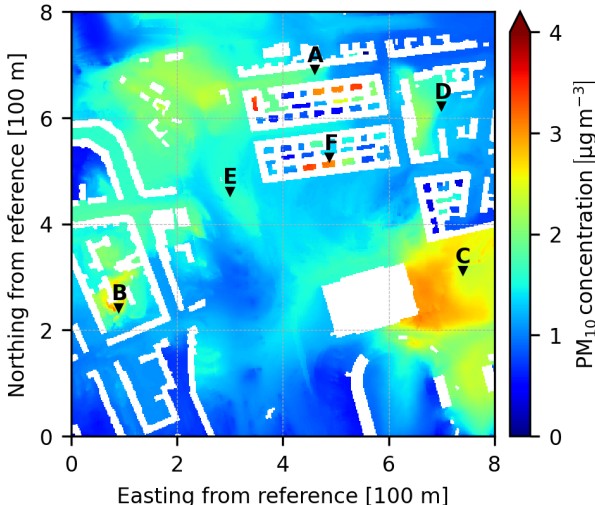

**Figure 7.** Horizontal spatial distribution of diurnal mean PM$_{10}$ concentration evaluated at 2 m above ground, with markers indicating positions of sampling locations listed in Table 4.



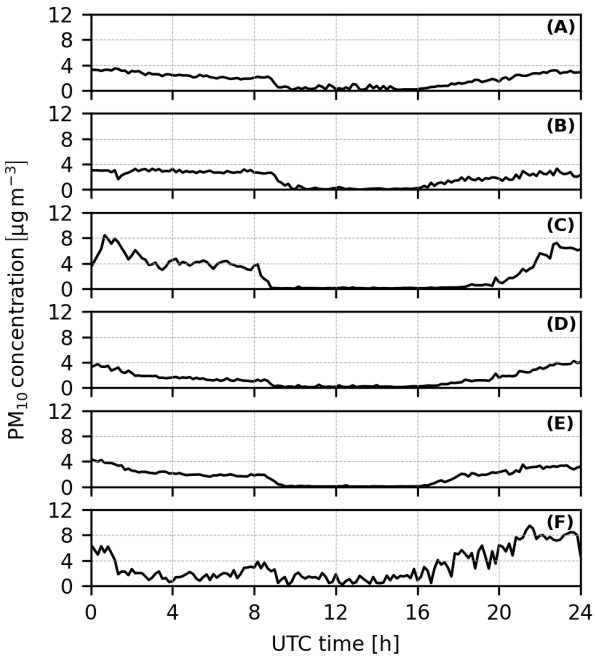

**Figure 8.** Diurnal time series for $PM_{10}$ concentrations at 2 m above ground for each sampling location.



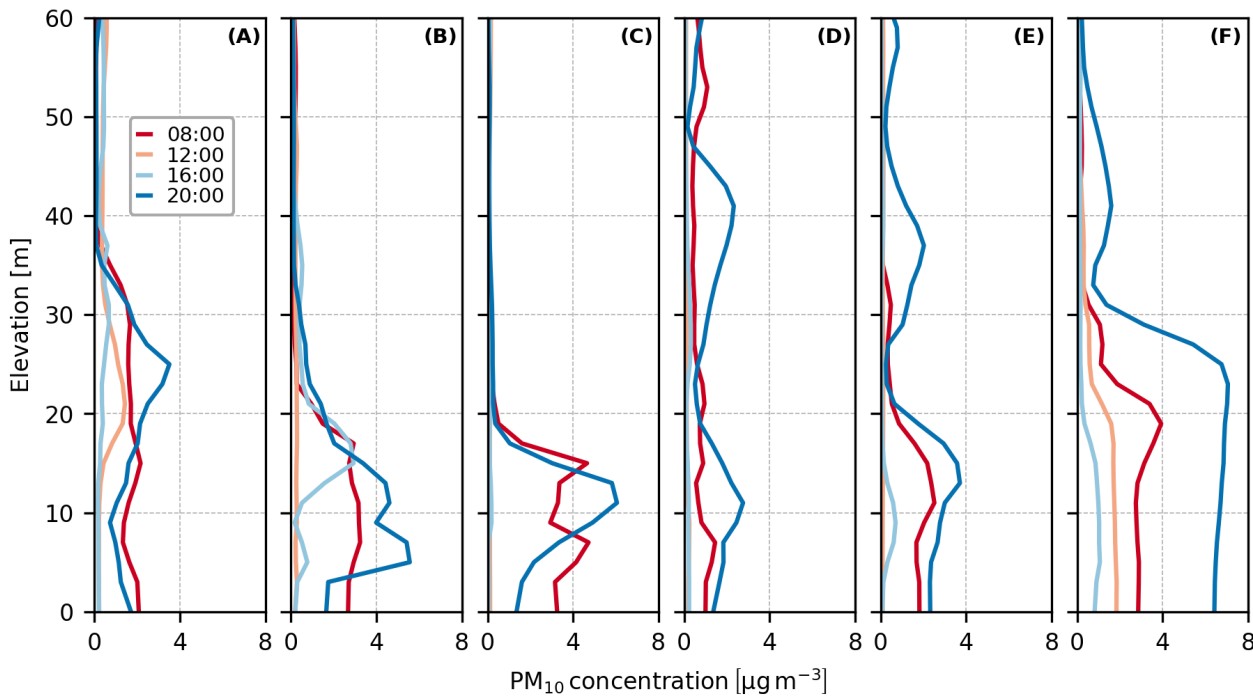

**Figure 9.** Diurnal mean vertical profiles for $PM_{10}$ concentrations above ground level for each monitoring location.




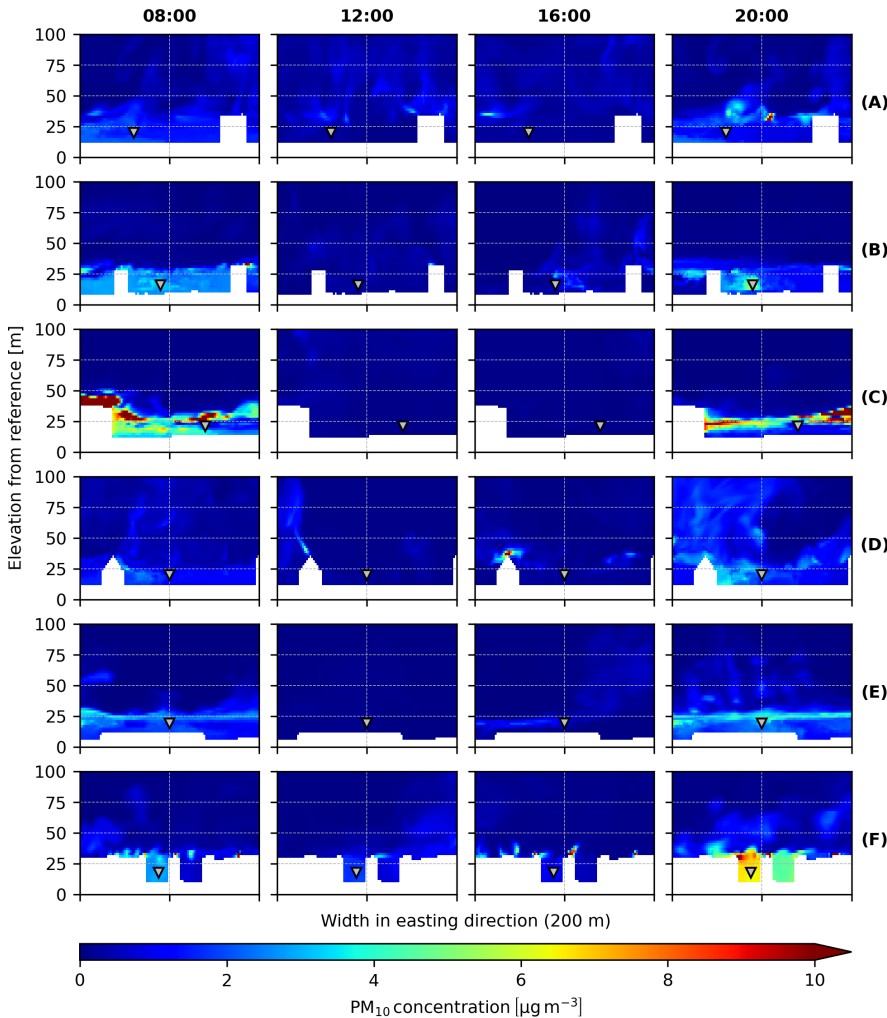

**Figure 10.** Vertical cross sections of $PM_{10}$ concentration in the vicinity of each sampling location at 08:00, 12:00, 16:00, and 20:00 UTC. The sampling plane is taken at the north-south position of each location. The east-west direction of each figure covers a distance 200 m. Triangular markers indicate positions of the sampling locations. Elevations are relative to a reference value of 36.87 m above sea level. [NOTE TO REVIEWERS AND EDITORIAL STAFF - The width of this figure has been reduced from 17cm to 12cm so that its contents fit the page in manuscript mode.]





**Table B1.** Building type ($\beta$) definitions.

| $\beta$ | Function | Construction period |
|---|---|---|
| 1 | Residential | Pre-1950 |
| 2 | Residential | 1950-2000 |
| 3 | Residential | Post-2000 |
| 4 | Commercial | Pre-1950 |
| 5 | Commercial | 1950-2000 |
| 6 | Commercial | Post-2000 |

**Table B2.** Compactness factor ($\Phi_\beta$) and annual energy demand ($E_\beta$) for each building type ($\beta$). $\Phi_\beta$ is in the unit of m$^{-1}$, while $E_\beta$ is in kW h m$^{-2}$ per annum.

| $\beta$ | 1 | 2 | 3 | 4 | 5 | 6 |
|---|---|---|---|---|---|---|
| $\Phi_\beta$ | 0.23 | 0.28 | 0.28 | 0.26 | 0.29 | 0.29 |
| $E_\beta$ | 130 | 100 | 100 | 110 | 89 | 89 |

**Table B3.** Emission factors for major pollutant species for different domestic furnace technologies. CO and NO$_2$ are in the unit of mol TJ$^{-1}$, while PM$_{10}$, NO$_x$, and VOC are in kg TJ$^{-1}$.

| Technology | CO | NO$_2$ | PM$_{10}$ | NO$_x$ | VOC |
|---|---|---|---|---|---|
| Centralized oil | 0.1 | 2.1 | 0.34 | 45 | 0.5 |
| Centralized gas | 0.14 | 0.78 | 0.006 | 17 | 0.7 |
| Centralized wood pellets | 1.7 | 3.4 | 18 | 73 | 3.2 |
| Centralized woodchips | 1.6 | 4.2 | 27 | 91 | 1.8 |
| Centralized log | 8.3 | 3.9 | 40 | 84 | 22 |
| Wood stoves / fireplaces | 28 | 3.9 | 48 | 84 | 29 |