# Peer review of "An enhanced emissions module for the PALM model system 23.10 with application on $PM_{10}$ emission from urban domestic heating"

_Geoscientific Model Development, 2024_

## Author Comment (AC1)

Dear Editor:

Please find attached the Authors' response to the comments from the three reviewers on the following pages. In the mean time we would like to thank you for your efforts in organizing the discussion of this manuscript.

Please note that, in the current version of the MS, the width of Figure 10 has been shrunk from 17 cm to 12 cm so that it will completely fit on the page in manuscript move. Once accepted for publication, Figure 10 should revert to 17 cm for better visualization.

Sincerely yours,

The Authors.

**Response to Reviewer 1**

The Authors would like to thank Reviewer 1 for the encouraging comments. Please find our response on the corresponding comments below, with reference to the Reviewer's original comments in indented italics.

> **Comment 1:** *Page 3: Equation 1: PM10 particles contain large particles with diameter > 5 micrometers. Such large particles can experience gravitational settling, which requires another term in the transport equation (1). I would say the transport equation applies to smaller particles within PM2.5 that do not experience gravitational settling. Is it fair to say that the modeling approach is more suitable for PM2.5 than PM10? If you agree, then the model should be advertised as a transport model for PM2.5. If larger particles are to be accounted for correctly, then a gravitational settling term should be added to the transport equation.*

**Response 1:** The PM emission factors for domestic heating from Struschka and Li (2019) is only applicable to $PM_{10}$ and is accordingly indicated throughout the MS.

However, the Reviewer is correct, that gravitational effects (i.e., settling) may alter the dispersion characteristics, and can be activated through PALM dry deposition module (Khan et al, 2021). This has not been considered in the MS for two reasons. First, this MS is primarily a model description of an extended, generalized approach for emissions source treatment in the prognostic equations, and as such the application to domestic heating serves as a demonstration of said approach. Second, the bulk and turbulent flows are frequently the dominant modes of particulate transport. A note has been introduced at the end of § 3.1 to indicate that gravitational effects have not been considered.

> **Comment 2:** *Section 3: Particles emissions from domestic heating are only present when fossil fuels (e.g. natural gas or diesel) are used which result from combustion processes in HVAC furnaces and water heaters. However, many buildings rely on heat pumps for heating, which only consume electricity. In such a case there will not be particle emissions from heating. Does PALM allow for such heating technologies? If so, particle emissions from heat pumps should be set to zero.*

**Response 2:** Yes, under LOD2, the user has a much greater degree of control over the location, mode, and the amount of pollutants emitted into the solution domain. This has been indicated in §§ 2.3.1, 2.3.3, and 3.2 of the MS. Further, if the building in question employs heat pump, it is expected the emissions would originate from the power plant(s) where the electricity is generated, assuming coal-fired or gas, instead of the building stack directly. This results in a displacement of both the geographical location of the emission source, as well as the assignment of emission GNFR sector C to GNFR sector A. Quantifying the corresponding change in emission inventory is conceivable but is beyond the scope of this MS.

> **Comment 3:** *Equation 12: This equation is used to consider heating requirement for buildings as a function of $T_{outdoor}$-$T_{setpoint}$, however, I think it is more appropriate to use heating degree days instead. Even in the absence of a building energy system, $T_{indoor}$ can be higher than $T_{outdoor}$ due to internal heat gains (people, equipment), conductive-convective-radiative heat transfer, and thermal inertial of buildings. An energy balance in the absence of building energy systems results in degree days. Would it be more appropriate to estimate the building heating needs using the degree-day approach, rather than $T_{outdoor}$ – $T_{setpoint}$ one can use $T_{outdoor}$-$T_{indoor}$ in the absence of building energy systems? In summary, fundamentally, degree-days ($T_{outdoor}$-$T_{indoor}$) estimates are different from temperature deficits ($T_{outdoor}$-$T_{setpoint}$). Weather and climate models calculate degree-days considering an*

*energy balance model between buildings and the outside environment in the absence of a building energy (heating/cooling) system.*

**Response 3:** The parameterized domestic heating module uses a generalized form of heating degree day (HDD) to characterize emissions at a much higher temporal resolution than once-on-a-daily basis. Since the definition of HDD can be effectively regarded as a deficit between the ambient and target temperature, this generalized approach is thus termed "temperature deficit", described in § 3.1 of the MS.

However, the Reviewer is correct. The indoor temperature should be calculated at each building. At the time of implementing the domestic heating emission parameterization, and subsequently the preparation of this MS, such module, while it exists for PALM (Fröhlich and Matzarakis, 2020), does not have a functional interface with the PALM chemistry module (Khan et al, 2021). Thus a set point temperature was introduced as a placeholder input, with a default value of 15 °C has been adopted according to guidelines provided by VDI (2013). The latter reference has been added to the MS.

**References:**

Fröhlich and Matzarakis (2020) GMD 13-3055-2020.
Khan et al (2021) GMD 14-1171-1193.
Struschka and Li (2019) "Temperaturabhängige zeitliche Disaggregation von Emissionen aus Feuerungsanlagen der Haushalte und Industrie für Berlin im Rahmen des MOSAIK-Projektes."
VDI (2013) VDI-Richtlinie 4710 Blatt 1.

**Response to Reviewer 2**

The Authors would like to thank Reviewer 2 for the meticulous comments. Please find our response on the corresponding comments below, with reference to the Reviewer's original comments in indented italics.

> **Comment 1:** *[I]t is strongly recommended to perform tests treating domestic emissions as either buoyant or nonbuoyant volume sources. The heating exhaust should be a warm plume, which rises by buoyancy, especially in winter when air temperatures are low. The buoyancy effects might be less effective than the turbulent mixing, however, this needs to be investigated for the conditions of this study. It is referred to a study by Langner and Klemm (2011), who demonstrated that dispersion models work acceptably for nonbuoyant volume sources, but don't cope with buoyant volume sources. Another aspect of PM10 emissions from domestic heating is that they are partly volatile. Residential emissions of organic carbon are largely semi-volatile and intermediate volatility compounds (S/IVOCs). The authors should explain how the modular emission concept can be extended in the future to handle the volatile fraction of emissions and incorporate the emissions of S/IVOCs. They should also discuss the representativeness of the meteorological conditions in the 48-hour simulations for the winter period.*

**Response 1:** Volatility of emitted particles has not been considered in this work due to performance constraints, as these calculations are computationally expensive. However, PALM integration with existing other aerosol models, that is SALSA 2.0 (Kurppa et al, 2019) and ISORROPIA (Nenes et al, 1998; Fountoukis and Nenes, 2007) are available, where the PALM chemistry module, and by extension the present emission module, can operate seamless with these models, should this be of scientific interest.

Buoyancy effects, otherwise referred to as plume rise, is currently not implemented in the PALM Model System. However, in the context of domestic stacks featured in § 3.2, they are often small and will disperse quickly into the surrounding atmosphere, which means explicit treatment of buoyancy effects are of secondary importance, though out of prudence they should not be overlooked, as pointed out by the Reviewer.

These aspects have been added to § 3.1 of the MS for emphasis.

> **Comment 2:** *Introduction (P3, line 58-61): The two examples (trees and exhaust emissions from aviation) given here do not have much in common. Which vertical resolution is meant in relation to trees and aircraft? Approaching and starting airplanes emit in a height up to 900 meters within several kilometers around airports, potentially affecting ground concentrations. Further, the phrase "sufficiently low horizontal resolutions" sounds strange, as models generally try to achieve high resolution.*

**Response 2:** The Reviewer has pointed out a very important point that led to the implementation of the emission module. On one hand, emission production mechanisms are vastly different, from biogenic, to road traffic, to aviation. On the other hand, treatment of the emission source terms at the prognostic equations is the same irrespective of production sources. Thus, one of the main design considerations for this emission module is to provide an efficient and uniform framework for the assignment of emission sources into the prognostic equations, while allowing the flexibility of taking into account the variability in emission mechanisms defined, for instance, in each emission sectors. How this aspect of the emission module has been implemented PALM is discussed in §§ 2.2 and 2.3.

On grid resolution, PALM is currently considered a high-resolution model. As such some techniques used in regional or global models no longer apply (i.e., definition of vertical emission profiles) as emissions can (and must) be explicitly assigned to individual cells. However, there is always a trade-off between detail

and performance, and as such model runs might need to be conducted at lower grid resolutions, say, 10 m or 50 m. This is still very high in comparison with regional or global models, but can be "sufficiently low" that a cell could include multiple emission sources from different sectors

> **Comment 3:** *Introduction (P3, line 64-65): Volume sources are a quite common way to treat diffusive sources in dispersion models. Mention how other models for the urban scale, e.g. AEROMOD and AUSTAL deal with (nonbuoyant) volume sources.*

**Response 3:** While the Authors agree with the Reviewer on the treatment of volume sources, PALM is a non-steady Eulerian model, while AEROMOD is a steady-state Gaussian dispersion model and AUSTAL is a Lagrangian dispersion model. As such, the treatment of volume source terms is thus quite different in each model, and, after some contemplation, the Authors have decided to refrain from referencing these two models, at the risk of misrepresenting familiarity in their usage and implementation, as well as implying any functional equivalence in source term treatment with PALM.

> **Comment 4:** *Model description (P5, line 123-125): While the hash map is described as a clear connection between the emission database and the cell coordinates (i,j,k) where the emission of a source is added to the prognostic equations, it is not clear what happens for different cell sizes and volumes of the defined grid. How is it assured that the emission source is allocated to the correct cell when the grid cell size and volume is changed in the model configuration?*

**Response 4:** In PALM, the prognostic equation (Eq 1) undergoes volume integration after the emissions have been assigned. Emission inputs will be adjusted to the required grid cell size without user intervention. Volume integration is standard in models based on the finite difference / volume approach and therefore is not explicitly mentioned in the MS.

> **Comment 5:** *What is the footprint of a building and how is it calculated (P8, line 213)?*

**Response 5:** The footprint is the projected area of the building on the ground. This information is provided as input, as indicated in § 3.2. This information is typically available from the city's planning department as part of the GIS data. This has been clarified in § 3.1 when this term first appears.

> **Comment 6:** *Module implementation: It is not clear how the height level of the building stack is considered. The module implementation section only mentions the (i,j) cell location of each building stack. The volume source is probably defined at the height of the stack exit and not the entire building is the volume source. Are there any plausibility checks of the user-provided emissions? There should be some internal control in the emission modules that check the plausibility of finally calculated emission rates and gives warnings when emission rates are unrealistic or not defined.*

**Response 6:** For domestic parameterization (LOD 0), the stack volume sources will be introduced unconditionally in the cell above the roof of the corresponding building at horizontal stack location $(i, j)$. While there is no plausibility check during calculation of the volume sources, which would otherwise be quite time consuming, rigorous checks are implemented on the parameterized input to ensure the calculated volume sources are physically sound given the user input. On the other hand, under LOD 2, described in §§ 2.3.1, 2.3.3, and 3.2 of the MS, emissions sources are provided as external data, the onus is on the user to ensure correct specification.

> **Comment 7:** *It would be interesting to see a more generalized approximation of the vertical profiles shown in Figure 9 for sampling sites A-F as time average, for example in steps of 10 m above ground. The average vertical profiles should be compared to more generic vertical profiles of heating emissions in urban areas found in the literature.*

**Response 7:** As a model description paper, the case runs and corresponding results discussed in §§ 3.2 and 4, are conceived and presented to serve the sole purpose of demonstrating consistency of input and output data in the context of the model. Interpretation of the model output beyond verification of the functionality of said model, as well as any of their comparison with published data, is *ultra vires*.

> **Comment 8:** *Define the reference height (P16, line 476). What causes the vertical mixing of heating emissions, does buoyancy of the heating plume play a role here or not? The occurrence of down-wash and accumulation should be explained in terms of meteorological conditions, not only in terms of trapping in building enclosures.*

**Response 8:** The reference height (36.87 m above sea level) is calculated internally by PALM for computational reasons, as the topology of the solution domain is not flat, and cells lying underground, and those inside urban structures, will remain unused. The description of the reference height in § 4 has been expanded for clarification.

Vertical transport in open areas is mainly caused by the prevailing wind. In urban enclosures, the turbulent shear layer at the roof region restricts momentum, heat and mass exchange above and below the buildings (Bright et al, 2013; Chan and Butler, 2021; Driver and Seegmiller, 1985; Oke, 1988) which also attenuates the effects of ambient meteorology significantly.

However, the instance of "vertical mixing" indicated by the Reviewer has been corrected to "vertical transport" to indicate a more general transported mechanism than a purely meteorological concern, which is used throughout the MS.

> **Comment 9:** *Figure 10: in top row (A) the vertical cross-section shows a hotspot at 20:00 at around 30 m, despite there seem to be no emission stacks of buildings close-by.*

**Response 9:** The emission sources lie outside of the cutting plane in the figure in question. That the urban structures around sampling location (A) are not aligned with the cutting plane makes the visualization more difficult. A note has been provided in the last paragraph of § 4.3 to remind the readers of this.

> **Comment 10:** *The Concluding remarks should address the limitations of the domestic emission parametrization. The uncertainties of the emission factors are large and cannot be ignored. Also the diurnal variation in domestic heat usage can be locally different from the one defined in CAMS for other stationary combustion.*

**Response 10:** The CAMS profile used in this MS is the default configuration, but the user can introduce other profiles that are deemed more suitable for the particular model run. This is indicated in §3.2 of the MS. A more geographically refined emissions specification is also possible, either through LOD 2 or the generic emissions mode (§ 2.3.3). Clarification has been introduced in § 5 on the additional limitations of the parameterized domestic emissions module.

**References:**

Bright et al (2013) Atmos Environ 68 127-142.
Chan & Butler (2021) GMD 14 4555-4572.
Driver & Seegmiller (1985) AIAA J 23 163-171.
Fountoukis et al (2007) ACP 7 4639-4659.
Kurppa et al (2019) GMD 12 1403-1422.
Langner & Klemm (2011) J Air Waste Manage Assoc 61(6) 640–646.
Nenes et al (1998) Aquat Geochem
Oke (1988) Energ Buildings 11 103-113.

**Response to Reviewer 3**

The Authors would like to acknowledge the efforts of Reviewer 3. Please find our response on the corresponding comments below, with reference to the Reviewer's original comments in indented italics. In terms of language-related issues, they have all been addressed unless otherwise stated.

> **Comment 1:** *In general, what is meant by "hash map"? The relation in Eq (3) and (4)-(6), or an array (or other data structure) of emission sources indexed by kappa?*

**Response 1:** A "hash map", otherwise known as "hash table", is a fundamental data structure that provides a computationally efficient access to data, through the use of aptly called "hash keys". In the context of this MS, the hash map ($h^m$ or $H$) is a linear array which is associated with discrete emission sources in the computational domain using unique hash key values ($\kappa$) derived from their corresponding grid locations ($i$, $j$, $k$). Each emission module described in §§ 2.3.1 and 2.3.2 maintains its own hash map ($h^m$), as defined in line 100 of the MS, and subsequently Eq (2). The bidirectional relationship between the grid indices and hash key is defined in Eqs (3-6). A reference (Cormen et al, 2009) has been included in § 2.1 to provide additional background information for the interested reader.

> **Comment 2:** *What I miss is a description of the data structure of emission sources. How is it organized? For a given cell (i,j,k), how are the emission sources in this cell found? (i,j,k) maps to a single kappa value. How are the emission sources found then?*

**Response 2:** Please refer to Authors' response to Comment 1.

> **Comment 3:** *If the array of emission sources is shorter than the number of grid cells, there should be a search operation to find the sources corresponding to a given grid cell. If the array of sources has the same length as the number of grid cells (which makes look-up easy), it seems one could just use a full 3D field of source strengths instead, with the same memory cost but less complicated code.*

**Response 3:** As indicated in lines 92-93, and subsequently demonstrated in §§ 3.3 and 4, emissions sources are non-contiguous and, except for artificial test cases, only found in a small number of grid cells. This makes the use of the hash map approach critical, as it affords significant savings in terms of input storage and application memory. The search of the cell index is also accomplished by looking up the sorted hash keys in the hash map (see lines 188 and 199 of the MS).

> **Comment 4:** *A comment of how the implementation handles domain decomposition would be helpful. The list of emission sources is presumably prepared for each MPI process?*

**Response 4:** The Reviewer is correct. An additional sentence has been introduced at the end of the second paragraph of §2.3.1 to emphasize this point.

> **Comment 5:** *"3. The interface between the prognostic equation solver and the emission module should be implemented to allow only localized data access to prevent propagation of data corruption into other emission sectors." I don't understand this statement. Data corruption would be an error in the implementation. Isn't it an obvious design objective that the implementation should be error-free?*

**Response 5:** The Authors agree with the Reviewer in principle. Further, based on the Authors' practice in various development projects including the one outlined in this MS, having a software architecture that localizes code and data access will help isolation of source(s) of implementation error and subsequent deployment of corrective measures. Thus, while achieving an error-free implementation is the obvious goal, a sound data and code encapsulation strategy will help achieve this goal much more effectively.

**Comment 6:** *Eq (2): the notation feels unnecessarily convoluted, with the W sets with multiple indices. Additionally, in "W ∈ 0, 1, 2, · · · up to the corresponding upper bound N" presumably the set does not include N, but this is not clear from the formulation.*

**Response 6:** Eq (2) is the mathematical definition of the hash table as indicated in the Authors' response to Comment 1. An equivalent, graphical representation can be found in Figure 1. Given the complexity of the overall module design, it is in the Authors' opinion, that Eq (2) cannot be further simplified. Further, the range of *W* has been made to explicitly indicate "up to but not including" in the MS.

**Comment 7:** *The mapping in eq. (3) is quite trivial, just enumerating all the grid cells. Usually a hash function implies something more, e.g. that the output space is smaller than the input space (although this is no strict of formal requirement).*

**Response 7:** The Reviewer is correct that the output space (hash table size) can be smaller than the input space (the emission sources). This will result in hash key collision, however, which requires additional resolution strategies. Eq (3) is seemingly trivial due to the use of a Cartesian grid system in the PALM model framework. Another common approach is the so-called bitwise operation, as described in Teschner et al (2003) or, more classically, in Jenkins (1996), which are used in other model systems involving moving and/or unstructured grids. For the purpose of the MS, Eqs (3-6) provide a sufficient function without adding complexity. This point, as well as the additional reference, has been incorporated in § 2.1 of the MS.

**Comment 8:** *Eq. (4) is wrong, it should have a division not mod. Additionally, there is an implied rounding down after the divisions, which could be indicated with a floor function or with round-down vertical-bars-with-hooks symbols. I don't understand Eq. (8) or the explanation above it. Additionally something is missing in the sentence "...p is the union all emission sectors".*

**Response 8:** The Reviewer is correct, Eq (4) should be a `div( )` operation, instead of the `mod( )` operation that is currently being shown, and this has been corrected in the MS. Further, the functions `div( )` and `mod( )` are understood to provide integer outputs. As the inputs to these functions, as defined in Eqs (2-5), are subset of integers (i.e., whole numbers), there should be no further ambiguity in the form they are currently presented in Eqs (4-6) on the MS.

**Comment 9:** *Eq. (9) What's meant by the union of hash maps?*

**Response 9:** The union operation adds all values of the same hash keys across all hash maps. Referring this to "addition" imply each hash map contains identical sets of hash keys and thus are directly commutable, which is not true in this case. This has been clarified in § 2.1 of the MS.

**Comment 10:** *line 268: "specifies the annual cumulative temperature (in degrees) to be heated above the ambient temperature to the target temperature, with a default value of 2100 K." It's not obvious what annual cumulative temperature is. If it is something like degree days, the unit is wrong.*

**Response 10:** The Reviewer is correct. The unit of both heating degree day (i.e., temperature deficit) and annual cumulative temperature should be in degrees. All instances indicating otherwise have been corrected throughout the MS.

**References:**

Cormen et al (2009) "Introduction to Algorithms" MIT Press 253-280.
Jenkins (1997) Dr Dobbs J 22(9):107.
Teschner et al (2003) Proc 8 Int Fall Workshop Vision Model & Vis.

---

## Author Response (AR2)

Dear Editor:

Thank you for your efforts so far in managing the revision process thus far and ensuring the high quality of this publication. At this stage of the review, we are very encouraged by the positive feedback from Reviewers 1 and 2. The minor issues requested by Reviewer 2 has been fully addressed. Concerning Reviewer 3, there are a number of issues with the critiques of Reviewer 3 that we would like to bring to your attention. It is in our hope that, with your input, we could come up with a reasonable course of action.

In particular, the Authors believe that the critical comments from Reviewer 3 are predicated on a misunderstanding of interpreting the hash map approach as a verbatim replacement of a 3D array. The Authors further emphasize that this approach enables emission source to be represented only at the source location only, which offers significant savings in runtime memory and storage. Additional clarification has been introduced in § 2.1. And, along with various minor comments, the Authors hope that the amendments will be satisfactory, and that the overall implementation has been sufficiently detailed not to hinder reader's understanding.

In light of the above arguments, the Authors will proceed to disseminate the comments from Reviewers 2 and 3, and to address all necessary and outstanding issues. The original reviewer comments will be presented in indented italics, and abridged only to clarify the Authors' understanding. The line and section references to the manuscript refers to the first revision.

As emphasized in the previous discussions already, in the current version of the MS, the width of Figure 10 has been shrunk from 17 cm to 12 cm so that it will completely fit on the page in manuscript move. Once accepted for publication, Figure 10 should revert to 17 cm for better visualization.

Sincerely yours,

The Authors.

**Response to Reviewer 2**

> **Comment 1:** *Equation (10) assumes that 100% of the energy consumption is used for heating. In reality, a certain fraction of the energy is used for water heating (and cooking with gas) that does not depend on the temperature deficit.*

**Response 1:** A note will be added in §3.1 to bring this to the readers' attention. This change will be considered for implementation in future releases of this module.

> **Comment 2:** *Page 12, line 346: typo: "boarder".*

**Response 2:** This has been corrected.

> **Comment 3:** *Section 4: give a list of input data for the emission calculation that needs to be provided by a user to reproduce the exemplary model run on domestic heating emissions. Which checks of the input (in addition to minimum height and footprint) are performed to ensure that the calculated volume sources are physically sound?*

**Response 3:** The Authors have introduced Appendix C, indicating user-defined and default values for all namelist parameters used in the domestic model, as well as all relevant building-specific input data in the PALM `_static` file. Additional information on input and runtime checks have been introduced at the end of § 3.2. The implicit understanding here is that the model can only rectify faulty input and calculated values to the extent of preventing system crashes, usually by replacing it with default values during initialization, and imposing a zero lower limit in emission source values during runtime.

> **Comment 4:** *Page 15, line 460-461: rephrase wording of the sentence "This indicates mixing of the PM10 still lingering …".*

**Response 4:** "still lingering" has been replaced with "that remains".

**Response to Reviewer 3**

> **Comment 1:** *[T]he implementation makes the hash tables as large as the model's 3D grid ... Crucially, this offers no space savings over a simple 3D array.*

**Response 1:** The Authors believe there is a misunderstanding. The hash map approach was chosen so that only cells identified as emission sources are stored in the data structure, thus offering significant savings in runtime memory and storage. The motivations are stated in §2.1, LL 83-84, that the emissions sources "are only defined at sparse, discrete regions", and thus, in LL 89-90, "it is strongly preferred to consider [emissions] only at discrete locations where the emission source is present." The present implementation functions under the principle that the number of volume sources is not equal to the number of grid points.

At the suggestion of the Editor, the Authors have introduced an additional clarification in § 2.1 to justify the adaptation of the hash map approach over the traditional 3D array.

> **Comment 2:** *Equations (3) to (6) essentially implement indexing in a multidimensional array. I cannot see any reason why the re-implementation is preferable to just using a multidimensional array, which is very efficient in Fortran. The memory cost is the same, and the native Fortran implementation is easier and also faster, as the divisions and modulo operations in (4) to (6) can be avoided.*

**Response 2:** Please refer to Response 1 regarding the memory cost of the hash map, and Response 7 for the status of Equations (4-6).

> **Comment 3:** *What prevents me from recommending publication is the highlighting of the importance of using a hash function, followed by an implementation that offers no benefit but comes at the cost of both performance and clarity. Additionally, the implementation is still not explained clearly enough, see the minor points below where several unclear statements are noted. These flaws are especially severe for a journal such as GMD, focused on model implementation and a clear description of it. "ideally, the description should be sufficiently detailed to in principle allow for the re-implementation of the model by others, so all technical details which could substantially affect the numerical output should be described." (from point 3 of GMD's guidelines for Model Description Papers).*

**Response 3:** Please refer to Responses 1 and 2. In the first author's practice in computational model development, particularly in the private sector, hash tables are commonly used as a memory efficient method to represent sparse, discrete data. As such, the Authors are pleasantly surprised such classic concepts still find novel uses. On the other hand, as a fundamental data structure, there already exists an abundance of comprehensive resources on the theory and machinery of hash tables. The Authors thus add no scientific value to this article by deviating its focus from its application in organizing emission sources to a rudimentary exposition of its implementation, a topic that has been thoroughly covered in textbooks on computer data structures and algorithms, with Corman et al (2009) is one of the most accessible to the readers.

Having said that, aside from the indicated minor comments, no further details are given on the reason or manner in which the implementation is not satisfactorily explained. The Authors assume that addressing these comments will remedy any concerns from the Reviewer.

> **Comment 4:** *As a side note, when storing the emission sources in a simple 3D array, there is a simple optimization: Store the emission sources from the surface up to k_max, the height level of the highest emission source. For cases where sources are*

*located between the ground and the highest smoke stacks, this can already save a substantial fraction of the memory.*

**Response 4:** The Authors would also like to point out that, at lower grid resolutions (i.e., regional and global scale) models, increasing $k_{max}$ by one can cover the vertical dimension by tens to hundreds of meters. In urban scale models such as PALM, the same vertical space must be covered by 10 to 20, or even more vertical layers due to the high spatial resolution. At this stage, the memory consumption, while substantially reduced in the Reviewer's opinion, can still severely restrict model scalability and runtime performance.

> **Comment 5:** *Line 195: The sorting procedure alluded to here is very unclear. Why is it needed at all? Why is the reverse lookup not achieved through Eqs 4-6 this time? If it's a crucial part of the implementation it should be explained carefully, perhaps together with the global hash map H.*

**Response 5:** Line 195, and by extension §2.3.2, do not refer to any reverse lookup (see Response 7). However, the Reviewer is correct, the sorting algorithm (§2.3.2, L194) "facilitate[s] the hash key lookup" such that the runtime scales only with the logarithm of the number of emission sources using bisection search. This will be added to the MS.

> **Comment 6:** *Line 115, Equation 7: It is still not well explained what is stored in h^m (h with superscript m). I expect h^m to give a source intensity for each grid cell with an emission source from sector m. But it's also said that f depends on the location, making that interpretation seem redundant. There is a notation problem: if h^m is the mapping between (i,j,k) and kappa defined in Eq. (2), what does it mean to multiply h^m with f in Eq (7)?*

**Response 6:** The definition of $h^m$ is given in Eq. (2), in which (§ 2.1 LL 98-100) "the spatial association between the … emission source location and the corresponding … cell index locations in the computational domain are maintained … for which … a hash key κ is assigned for each source location[.]" The function $f_p^m$ (the source function) calculates the intensity of emission. The dot product operation ($h^m \cdot f_p^m$) in Eq. (7) is not redundant as $f_p^m$ is continuous (that is, it produces a value as long as there are input parameters), and the dot product implies that $e_p^m$ is only defined where $h^m$ is defined (i.e., at the respective volume sources). A short clarification will be added to L 116 to reinforce the idea that $e_p^m$ is only defined at volume source locations.

> **Comment 7:** *Line 110: "It should be noted that Equations (4 - 6) must be performed in the order presented." this seems to no longer apply to the revised version, where the three equations are independent.*

**Response 7:** After reviewing the source code, Equations (4-6) are not used in the emission module aside from model diagnostics. They, and the descriptions associated with them, have been removed from the MS to improve overall clarity.

> **Comment 8:** *Line 111: "or by using bitwise operations" it seems hard to guarantee the hash map is still collision free in this case, however that's a side point.*

**Response 8:** As a (side) point of interest, the Reviewer is correct; bitwise operations do not unconditionally guarantee collision-free hash keys as Eq. (3) does. In practice, the use of bitwise operation for unstructured and moving meshes has not (yet) resulted any hash key collision, at least for the Authors, in this application in other research and commercial model codes. It should also be added that the method proposed by Teschner et al (2003) is particularly effective in minimizing such collisions.

**Comment 9:** *Line 244: "using Equation (7) in the function The function f, as a function of time" something wrong in the sentence, maybe just a missing "."*

**Response 9:** The Reviewer is correct. L 244 should read "… hash key κ – using Equation (7)." And the remaining text should be removed. It will be corrected.

**Comment 10:** *Line 391: "4.1 Temnperature deficit" typo[.]*

**Response 10:** The Reviewer is correct. It will be corrected.

**Comment 11:** *Fig 4 and Table 2 caption: I'd suggest to omit empty unit brackets*

**Response 11:** The empty unit brackets emphasize that the terms in question carry no units. Their inclusion is also to introduce stylistic consistency with other caption labels.

**Comment 12:** *[T]he "2.1E-4" etc notation for powers of 10 doesn't look good in text. Also, the numbers could probably be given with fewer digits.*

**Response 12:** Prior to the initial submission, the Authors have experimented with different notations, and concluded that the "E" notation offered the least bad option in terms of space and clarity. The five significant digits used throughout the manuscript is to partially reconcile the vast difference in orders of magnitudes of the tabled values, particularly in Table 2.

**References:**

Cormen et al (2009) "Introduction to Algorithms" MIT Press 253-280.
Teschner et al (2003) Proc 8 Int Fall Workshop Vision Model & Vis.

---

## Author Response (AR3)

Dear Editor:

We appreciate your immense patience in handling the review process of this MS. After going through the remaining comments from Reviewer 3 thoroughly we have prepared a revised MS, as well as our response on the following page.

As we pointed out in previously, in the current version of the MS, the width of Figure 10 has been reduced from 17 cm to 12 cm so that it will completely fit on the page in manuscript mode. Once accepted for publication, the width of Figure 10 should revert to 17 cm for standardized visualization.

Sincerely yours,

The Authors.

**Response to Reviewer 3**

> **Comment 1:** *I guess [out lines the program flow of the PALM vsrc module] If this is the case, please say it in the article.*

**Response 1:** The general architecture of the emission source module has been described in §2.1. The concept has been outlined in LL 112-118. The definition of the hash map and corresponding hash keys are defined in Eqs (2-3). The implementation-specific details can be found in §2.3.2. As an implementation specific issue, the discussion of the sorting and subsequent bisection (binary) search is mentioned in §2.3.2. In the Authors' view, the algorithm behind the present emission module has been described in the necessary detail, and its implementation sufficiently specified without losing generality. To reiterate these points, LL 98-119 in §2.1 of the MS have been revised.

> **Comment 2:** *However, this is \*not\* what is generally described as a hash table ... The data structure used by the authors could perhaps be described as a binary search tree.*

**Response 2:** The data structure the Authors employed for the emission module is a hash table. It contains the following components:

1) An (1D) array containing the data, in this case the $\epsilon_p^m$ terms defined in Eq (1), and
2) A hash function, as defined in Equation (2), which effectives maps the cell index locations to a unique hash key.

Details on handling and implementation of collision detection have been left out in the description as the hash keys in this case are unique to each cell point and thus unconditionally collision-free. The revisions made in Response 1 also emphasize the uniqueness of the hash keys.

While the Reviewer pointed out differences between the Authors' implementation of the hash table and what are generally described in textbooks, it is very common practice to store hash keys and their associative array indices in memory instead of calculating them on-demand for performance.

Further, in response to the Reviewer's claim on binary search trees, a binary (search) tree is a hierarchical data structure that an array used here is not. To expedite discussion, the rudiments of binary tree can be found in §2.3.1 of Kunth (1997) and Chapter 12 of Cormen et al (2022).

> **Comment 3:** *The addition "representing the number of emission sources." on line 124 does not seem to make sense here, as the N:s in (2) are grid sizes in i,j,k. N_kappa^m might be the number of emission sources.*

**Response 3:** The phrase "representing the number of emission sources" has been removed.

**References:**

Cormen et al (2022) "Introduction to Algorithms" 4 Ed MIT Press 312-330.
Kunth (1997) "The Art of Computer Programming" 3 Ed Addison Wesley Longman 308-330.